# RETHINKING CROSS-LINGUAL GAPS FROM A STATISTICAL VIEWPOINT

## ABSTRACT

Any piece of knowledge is usually expressed in one or a handful of natural languages on the web or in any large corpus. Large Language Models (LLMs) act as a bridge by acquiring knowledge from a *source* language and making it accessible when queried from *target* languages. Prior research has pointed to a *cross-lingual gap*, viz., a drop in accuracy when the knowledge is queried in a target language compared to when the query is in the source language. Existing research has rationalized divergence in latent representations in source and target languages as the source of cross-lingual gap. In this work, we take an alternative view and hypothesize that the variance of responses in the target language is the main cause of this gap. For the first time, we formalize the cross-lingual gap in terms of bias-variance decomposition. We present extensive experimental evidence which support proposed formulation and hypothesis. We then reinforce our hypothesis through multiple inference-time interventions that control the variance and reduce the cross-lingual gap. We demonstrate a simple prompt instruction to reduce the response variance, which improved target accuracy by 20-25% across different models.

## 1 INTRODUCTION

Large Language Models (LLMs) have revolutionized information access. Central to LLM's mission is to assimilate knowledge universally and make it available generally without any barriers. State-of-art LLMs are multilingual: Gemini supports over 40 languages (Gemini, 2025), GPT-5 supports at least 12 languages (GPT, 2025) (with no official number of supported languages) and open-source models like Gemma-3 support over 100 spoken languages (Gemma, 2025). Because pretraining data cannot contain duplicate information for every language, cross-lingual generalization is a necessary capability for LLMs. However, LLMs are known to have disparity in recalling knowledge across languages (Jiang et al., 2020; Kassner et al., 2021; Qi et al., 2023; Chua et al., 2024a; Goldman et al., 2025).

Our objective is to understand the causes of poor transfer of knowledge encoded in parameters across languages. We, therefore, evaluate models on knowledge-intensive tasks in a closed-book QA setting, i.e., without access to such tools as grounding in search. Cross-lingual gaps are quantified through disparity on parallel datasets that alter language-specific surface form of the prompts. We refer to the two evaluation settings that makeup parallel data as source and target. Prompts in the source split are (roughly) in-distribution with pretraining data, while those in the target split are out-of-distribution. Parity between source and target is achieved when the model generalizes across languages. For instance, consider a question derived from a Wikipedia article that is only available in Hindi: *When was Kreeda Bharti established?* Gemini[1] correctly answers the question when posed in Hindi (source) but the same question in Hebrew (a target language) is often answered incorrectly (Figure 1 (c)). LLMs are more likely to emit incorrect responses in target, which is the subject of our paper.

Figure 2 presents source-to-target performance drops on two recent benchmarks across various LLMs. Why do we see such cross-lingual gaps despite the mounting evidence (Dumas et al., 2024; Schäfer et al., 2024; Brinkmann et al., 2025) for language-agnostic representations in LLMs? Commonly, the problem is attributed to subpar generalization of parametric knowledge across languages

---

[1]Gemini-2.5-Flash (with thinking)

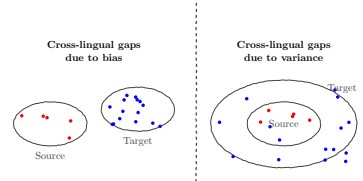
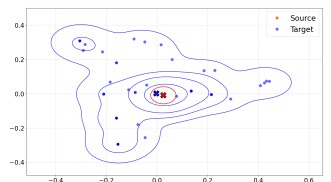
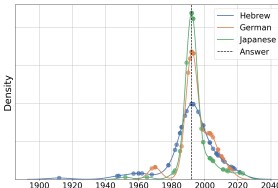

(a) Illustration of response distribution when the Cross-lingual gaps are due to bias or variance.

(b) PCA projection of responses (G-2.5-Flash) in source and target for the English-sourced question: *What was the name of the protagonist in the Prizzi novels?*

(c) Distribution of hundred responses (with G-2.5-Flash) for the Hindi-sourced question "When was Kreeda Bharti established?" in various languages.

Figure 1: If the model has knowledge barriers, we expect the responses in target biased as shown in left sketch of (a). In practice, we observe the target responses are distributed around source in such a way that their respective average values coincide as shown in the right sketch of (a). In (b), we show PCA projection of source and target responses where each response is a dot, and crosses represent centroid. We show KDE fitted distribution for three target languages for a numerical question in (c). Please refer Figure 14, 15 of Appendix for additional plots, and Section 1 for additional context.

due to representation misalignment. For instance, we may imagine knowledge fragmentation if Nelson Mandela and नेल्सन मंडेला (*Nelson Mandela* in Hindi) are embedded differently. Given the sparsity of non-English languages and non-latin scripts in pretraining data and even further rarity of entities, the representation misalignment rationale is compelling and likely. But could there be other causes at play? How can we validate if knowledge is failing to transfer?

Cross-lingual divergence in responses may emerge either due to variance (of responses) or biases. The well-known error decomposition of mean squared error, MSE = bias$^2$+variance, can be applied to our setting to characterize how differences arise between source and target responses. We contextualize bias and variance with an example question sourced from a Hindi document: *When was Kreeda Bharti established?* with the correct answer *1992*. If there is a knowledge barrier and the entity *Kreeda Bharti* is unrecognized in any language other than Hindi, we then expect the model to respond with a random guess anywhere from 1500 BC to 2024 AD leading to significant *bias* between target responses and the Hindi response (1992). On the other hand, if the gap is due to variance alone we expect target responses distributed more widely around the source response: say 1992±30. Figure 1 sketches source and target responses for a hypothetical example when the cross-lingual gaps are due to biases (left diagram) or variance (right diagram).

The distinction between the two sources of gap is important for guiding mitigation approaches. Besides, the problem is more severe if the gaps are due to biases because it requires rethinking LLM pretraining, tokenization, embeddings, etc. To the best of our knowledge, past work did not establish the nature of gaps. In fact, the literature often overlooked variance to explain the cross-lingual gap. The gaps instead are explained through certain knowledge barrier inducing biases in the target responses (Chua et al., 2024a; Wang et al., 2024b).

In contrast to the prevailing wisdom of attributing gaps to biases (i.e., knowledge barriers), we do not observe significant bias in the two examples of Figure 1(b, c). In (b), we show PCA projected response embeddings for a question sourced from an English document: *What was the name of the protagonist in the Prizzi novels?* with answer: *Charlie Partanna*. Please see Appendix C for details on how we obtained the embeddings. In Figure 1 (c), we sketched the various responses to our running example: *When was Kreeda Bharti established?* in three languages. The question is answered correctly in Hindi with high confidence, which we did not plot to show the variance in other languages. We highlight two observations based on Fig 1,( and 14, 15 in appendix), (1) the response distribution in target has higher variance, (2) the average of all responses coincides with the source response despite high variance.

Anecdotal examples of Figure 1 indicate variance over bias as the dominant cause of cross-lingual gap, which we establish carefully in the paper. In Section 2, we formally express the response distribution and cross-lingual divergence due to bias or variance. We report on experiments that

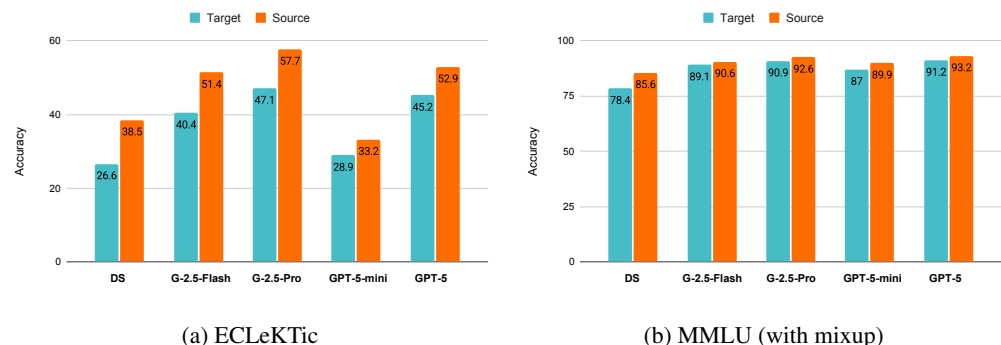

(a) ECLeKTic (b) MMLU (with mixup)

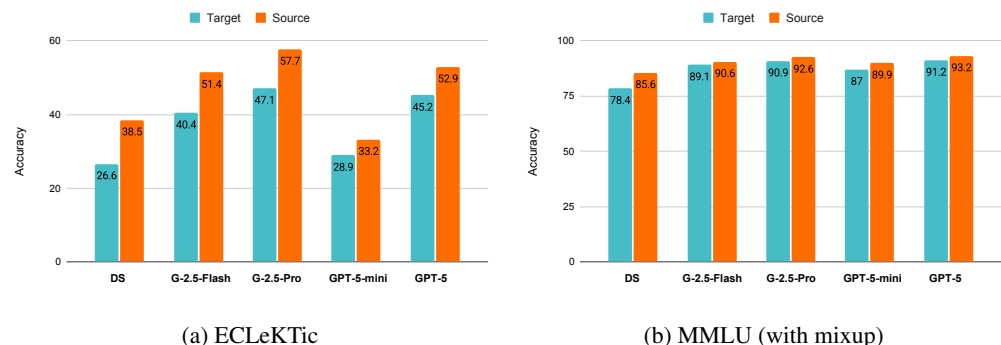

Figure 2: Cross-lingual performance gaps on ECLeKTic and MMLU (with mixup) with different LLMs. Here, DS refers to the DeepSeek model. G-2.5-Flash, G-2.5-Pro are Flash and Pro flagship Gemini models from 2.5 class and GPT-5-mini & GPT-5 are Open AI models. All models exhibit a significant cross-lingual gap on the ECLeKTic benchmark. Please refer Section 1 for additional context and Section 3 for dataset description.

tease apart the nature of gaps in Section 4.1. We discuss further implications of our findings in Section 4.2 including the surprising finding that cross-lingual gaps diminish when the model is confident in source. We finally conclude with takeaways, limitations and future work in Section 6. We hope an improved understanding of the source of gaps will better guide the mitigation efforts.

**Contributions:**

- We take an alternate viewpoint to explain cross-lingual gaps to hypothesize and validate for the first time that the gaps are dominantly due to variance (and not knowledge) transferring poorly from source to target (Section 4.1).
- We additionally demonstrate that (response) variance in Source and Target are proportional. As a consequence, source-target gaps diminish with decreasing variance in Source (Section 4.2).
- We validate our claims across two benchmarks and five closed/open SoTA LLMs. We present multiple inference-time interventions that mitigate the cross-lingual gap (Section 4.1).

## 2 A Framework of Cross-lingual gaps

In this section, we express how the target response distribution transforms due to bias or variance. We will begin by modeling the distribution of source and target responses by identifying the various sources of uncertainty. We ideally require access to model parameter posterior for characterizing the strength of bias-variance components, which in other words require training multiple LLMs on the same pretraining data. Since training an LLM, let alone multiple, is prohibitively expensive we model uncertainties in the forward pass for a fixed model. We operate under the assumption that floating point errors and MoE routing uncertainties sufficiently capture the response variance. If the forward pass of a model is deterministic, we would require training multiple models for bias-variance analysis. Thankfully, forward pass stochasticity sufficiently capture the response variance for the models we considered in our work.

An LLM model M projects an input $\mathbf{x}$ to its logits $\vec{z}$. Mapping from $\mathbf{x} \rightarrow \vec{z}$ could be stochastic due to floating point errors and MoE related routing variance. We model all those aspects by assuming the logits are sampled from a normal distribution with latent variables: mean $\vec{\mu}$, and variance $\sigma^2 I$. If the mapping from example to logits is deterministic, we can simply set the variance ($\sigma^2$) to 0. LLMs finally sample a response $\hat{y}$ via a categorical distribution parameterized by softmax transformation of logits. The sampling process is summarized below.

$$M(\mathbf{x}) \triangleq \vec{z} \sim \mathcal{N}(\vec{\mu}, \sigma^2 I)$$
$$\hat{y} \sim \text{Categorical}(\text{softmax}(\vec{z}))$$

The variance of responses is high if $\sigma^2$ is high or softmax$(\vec{z})$ is flat (i.e., has high entropy). Therefore, we may model increased variance of responses in target through multiplicative factors that increase variance or make the logits flatter.

For the sake of analysis, we assume that the response space is enumerable and shared between source and target. We may achieve this by collating many responses in both source and target and normalizing the unique values to only encode concept while ignoring the language. For instance, we normalize {*order of santiago, order de santiago, ordem de santiago*} to *order of santiago*. Hereafter, we will treat response space categorical with only the levels defined by the normalized values and logits are their corresponding scores.

With the response space normalized, we can quantify the probability of shared responses between source and target in the wake of response uncertainties. However, we must first model how the logit distribution tilts from source to target. We denote the mean and variance of logit distribution for source with $\vec{\mu}_s, \sigma_s^2 I$ respectively. The target logit distribution is expected to be unrelated to source indicating a bias if there are knowledge barriers, i.e., distributed with mean and variance $\vec{\mu}_b, \sigma_b^2 I$ respectively such that their modes do not match, i.e., $\arg\max \vec{\mu}_s \neq \arg\max \vec{\mu}_b$. If there is no bias but high variance, the target responses are expected to be distributed with a flatter logit mean $(\vec{\mu}_s/\tau)$ and higher variance: $\eta\sigma_t^2 I$ for some values of $\tau \geq 1, \eta \geq 1$. Since we do not know the relative contribution of bias and variance to cross-lingual gap, we model target responses as a mixture of both the distributions with an unknown mixing coefficient: $\pi, 0 \leq \pi \leq 1$. Overall, our model of source and target responses is summarized below.

Source response distribution:

$\vec{z} \sim \mathcal{N}(\vec{\mu}_s, \sigma_s^2 I)$.

$\hat{y}_s \sim \text{Categorical}(\text{softmax}(\vec{z}))$.

Target response distribution: $\eta, \tau, \pi$ as defined above

$\kappa \sim \text{Bernoulli}(\pi)$,

$\vec{z} \sim \kappa \underbrace{\mathcal{N}(\vec{\mu}_s/\tau, \eta\sigma_s^2 I)}_{\text{high var. component}} + (1-\kappa) \underbrace{\mathcal{N}(\vec{\mu}_b, \sigma_b^2 I)}_{\text{high bias component}}$.

$\hat{y}_t \sim \text{Categorical}(\text{softmax}(\vec{z}))$.

Figure 1 (a) depicts the two scenarios of cross-lingual gap. In the left sketch, the cross-lingual gaps are due to target bias, i.e., $\kappa = 0$. In the right sketch, the cross-lingual gaps are due to variance, i.e., $\kappa = 1$. Our objective in this work in a nutshell is to find the value range of $\pi$. Thankfully, the two mixture components have different expected behavior that we could establish the dominant component through few targeted ablations, which we describe in the rest of this section. Proofs for all the results can be found in Appendix D.

## 2.1 BIAS-VARIANCE DECOMPOSITION OF CROSS-LINGUAL GAPS

In this section, we study the nature of source-target gaps induced by the two components. Specifically, we discuss how the likelihood of source-target agreement transforms with reduced "response variance".

**Source-target gaps due to biases: knowledge did not transfer ($\kappa = 0$).**

**Proposition 1.** *When knowledge did not transfer, the probability of shared response between source and target decreases with decreased response variance.*

**Source-target gaps due to variance: confidence did not transfer ($\kappa = 1$).**

**Proposition 2.** *When the target responses are unbiased, the probability of shared response between source and target increases with decreased response variance.*

**Decoding the nature of gaps.** From Propositions 1, 2, we observe that the bias and variance components respond differently to response variance reduction. We may also interpret this intuitively from Figure 1 (a). Reducing the radii ($\sqrt{\text{variance}}$) will make the responses from source and target agree more often only when there are no biases.

In practice, we may reduce variance by simply ensembling multiple responses for the same example and using the majority voted response from N responses (Hastie, 2003). We validate if ensembling improves overall source-target agreement in Section 4.1. Decreasing source-target gaps with ensembling indicate that the gaps are dominated by variance and not bias, i.e., $\pi > 0.5$. We may estimate $\pi$ more accurately by estimating the fraction of examples on which ensembling reduced source-target gaps, which we discuss in more detail in Section 4.1.

## 2.2 FURTHER IMPLICATIONS OF UNBIASED NOISE IN TARGET

In this section, we discuss two surprising implications of the unbiased noise component: (a) variance in source and target are proportional, (b) cross-lingual gaps diminish with low variance in source. We also empirically validate our claims thereby further bolstering our main claim that knowledge barriers are not dominant.

We begin by showing that the response variance in source and target are related, and we quantify response variance using probability of the mode. We refer to probability of a sample matching the mode as confidence and represent variance as 1 minus confidence. For instance, when the response variance is 0, all the sampled responses match the mode with probability 1. We present the lower bounds on the confidence from source and target below using the notation from Proposition 2 and Section 2 for $\vec{\mu}_s, \sigma_s$.

**Proposition 3.** *Recall the sampling process when $\kappa = 1$. We have the following lower bound on the probability of the mode of responses sampled from source and target.*

$$\Pr(y_s^{mode}) \geq \left\{ \Phi\left(\frac{\mu_0 - \mu_1}{\sqrt{2(\sigma_s^2 + 2)}}\right) \right\}^{m-1} \quad ; \quad \Pr(y_t^{mode}) \geq \left\{ \Phi\left(\frac{\mu_0 - \mu_1}{\tau\sqrt{2(\eta\sigma_s^2 + 2)}}\right) \right\}^{m-1}.$$

*Where m is the size of the response space and $\mu_0, \mu_1$ are the top two values of $\vec{\mu}_s$.*

We make the following observations based on the above result.

1. When the source confidence is high, i.e., $(\mu_0 - \mu_1)/\sqrt{\sigma_s^2 + 2} \gg 1$ then the target confidence must also be high based on Proposition 3.
2. Since source and target confidence are related, we should see increasing agreement (or suppressed cross-lingual gap) as confidence in source increases.

We empirically validate the two observations in Section 4.2.

## 3 EXPERIMENT SETUP

**Datasets:** We employ two recent benchmarks: (1) ECLeKTic, (2) MMLU (with mixup).

- **ECLeKTic** (Goldman et al., 2025) dataset constitutes factoid questions sourced from Wikipedia pages that exist only in single language. Original questions from single language page define the *source* split. Translation into any other language make up the *target* split.
- **MMLU (with mixup)** (Chua et al., 2024a) builds on multiple-choice MMLU to introduce new examples that mixup the language of question and options randomly. Original questions make up the source split because examples with shared language for both question and options are likely in-distribution with pretraining. The mixedup questions make up the target split.

All our experiments pertain these two benchmarks. Both the datasets are knowledge-intensive and require recall from entities in foreign language/script. In Appendix H, we extend some of our results to Multiloko dataset (Hupkes and Bogoychev, 2025).

**Languages:** ECLeKTic, MMLU (with mixup) represent twelve, and five languages respectively. Please see Appendix E for further dataset details.

**Models:** We validate with five SoTA LLMs. For closed-source, we picked from Gemini (Comanici et al., 2025) series or GPT series(Jaech et al., 2024; Achiam et al., 2024). As a representative open model, we use Deepseek-R1 (Guo et al., 2025).

**LLM-as-judge.** Responses on ECLeKTic are freeform text, which we rate using LLM-as-judge. The LLM-judge rates if the model's response matches reference, see Appendix M.3 for the autorater

prompt. Autorater responses have over 95% accuracy on spot-checking but have slightly higher error rate on non-English reference/responses. For better validation of our claims without the influence of autorater noise, we use a split of ECLeKTic that require year as an answer, which we refer to as **Year-ECLeKTic**. Questions of the kind *In which/what year was . . .* make up about 18% of ECLeKTic. On Year-ECLeKTic, we use regex to extract year from reference and response and use exact match to mark the response as correct/incorrect.

## 4    EXPERIMENTAL RESULTS

We present results that support our contributions in this section.

- In Section 4.1, we show that reducing response variance through various ensembling approaches reduces cross-lingual divergence. The empirical result together with the analysis of Section 2.1 establishes that *cross-lingual gaps are due to variance (and not knowledge) transferring poorly*.
- In Section 4.2, we demonstrate that cross-lingual gap diminish with decreasing response variance (or increasing confidence) in Source. The section also validates the expectations from Section 2.2 if noise in target is unbiased.

### 4.1    CROSS-LINGUAL GAPS ARE DUE TO VARIANCE

In Section 4.1.1, we prompt the same example multiple times to ensemble responses, and in Section 4.1.2, we report on ensembling within a single prompt. Our intent in the section is to show improvements for each LLM without hinting any additional knowledge and by only reducing the response variance. So, all the ensembles are within the same model.

#### 4.1.1    RESPONSE ENSEMBLING

We sample ten responses per example for any LLM at default temperature with randomized seed for each prompt. We measure the divergence between "average" target and source response as the ensemble size increases from one to ten. We may better suppress the response variance with temperature set to zero instead of default temperature. However in practice, we noticed ensembling at default non-zero temperature reduced the variance better. Please see Appendix B for more details.

**ECLeKTic.** In Figure 4 (a), we plot the average distance between responses from source and target responses to the same question as a function of ensemble size. Since responses in ECLeKTic task are free-form text, we embed the responses using `text-multilingual-embedding-002` (Vertex AI, 2024) model from Vertex AI (AI, 2025). For each question, we compute the L2 distance between average embedding of responses from source and average embedding of responses from target (which includes eleven languages). We finally average L2 distance across all questions.
**Oracle.** We also plot the best possible L2 distance along with confidence region as the oracle in the plot. We estimated the oracle value as the average L2 distance among responses (to the same question) that are marked as correct by the LLM judge thereby ensuring their equivalence. The oracle value is non-zero due to embeddings encoding language and syntactics of text.
$\pi$ **estimate.** We present in each plot the value of $\pi = \mathbb{E}[\kappa]$ estimated as the fraction of examples on which the average L2 distance decreased from ensemble size of one to ten. Thinking-heavy LLMs (G-2.5-Pro, GPT-5, Deepseek) have slightly lower $\pi$ estimate likely due to their higher syntactic variation in responses that the embeddings did not fully suppress.
Results on Year-ECLeKTic in Appendix G further validate our claims on ECLeKTic without embedding-related artifacts.

**MMLU (with mixup).** In Figure 4 (b), we plot the average Chi-squared distance (described in Appendix M.4 for reference) between the probability distribution over the options. Recall that MMLU (with mixup) is a multiple-choice task and the answer is one of four options.
$\pi$ **estimate.** We present in each plot the value of $\pi$ estimated as one minus fraction of examples with mismatched predictions even after ensembling. We approximate binary mismatch with a soft score (Appendix M.5) with difference of mode probabilities. In Appendix M.6, we elaborate on why we needed different $\pi$ estimators for ECLeKTic and MMLU (with mixup) owing to their distinct response space: continuous vs categorical.

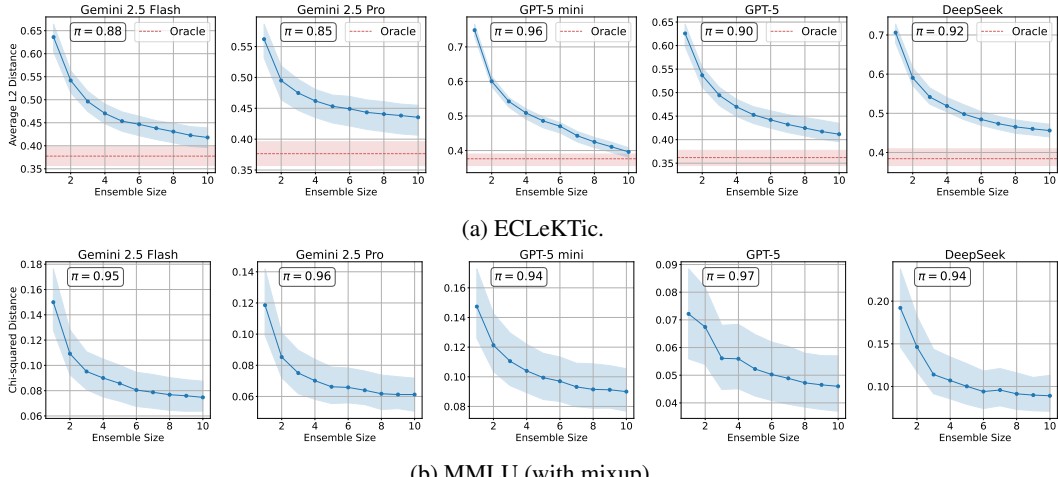

(a) ECLeKTic.

(b) MMLU (with mixup).

Figure 4: Response ensembling from multiple forward passes gradually diminishes the source-target differences as illustrated on ECLeKTic (top) and MMLU (with mixup) (bottom). Oracle value shown in red in (a) is the best expected value. Each plot shows an estimated value of $\pi$. Please refer Section 4.1.1 for details.

**Observations.** We observe a steady decrease in Source-Target divergence with ensemble size for both the benchmarks in Figure 4. The estimated values of $\pi \approx 0.9$ for ECLeKTic and $\approx 0.95$ for MMLU (with mixup) indicate that the noise is unbiased, i.e., $\kappa = 1$ for 90-95% of the examples. It may be possible that the remaining 5-10% examples may reduce further with even higher ensemble size or are biased (likely due to translation errors detailed in Appendix M.7 or due to cross-language factual inconsistencies). In Appendix K, we analyze cross-lingual transfer at a finer level. In the Appendix section, we show that the trend observed in this section holds for any combination high or low resource language transfer.

### 4.1.2 INPUT ENSEMBLING

A heuristic but popular alternate approach to ensembling is averaging model response across semantically similar inputs. Averaging responses with multiple inputs is shown to be as effective or better than ensembling with one input but multiple models (Kimura, 2021). Ensembling with semantically similar inputs is known as test-time augmentations (Shanmugam et al., 2021; Moshkov et al., 2020; Ayhan and Berens, 2018; Krizhevsky et al., 2012) and was found to be effective for improving estimates of predictions, robustness and uncertainty in various tasks such as image classification, segmentation.

TrEn-k is inspired from test-time augmentations. We prompt the model with k+1 semantically equivalent questions and elicit a single response under the assumption that the model implicitly ensembles across different questions. TTA is a variant on TrEn that forces the model to pay attention to all the translations (inputs).

Since the augmented questions look different, we may simply present all the (semantically equivalent) questions in one prompt and request an answer. The prompting is such that the answer needs to tally with all the presented questions, so we assume the model implicitly ensembles responses. We discuss two baselines in the spirit of input ensembling.

**Ensembling in prompt.** We introduce an ablation called **Translation Ensemble (TrEn-k)** where we present the original question along with k translations and then prompt for the answer. An example is shown in Figure 5 (a). In the appended translations, we ensure that we do not sample from the same script as the source. For example, if we are prompting a German-sourced question in Hindi, we do not sample translations from Hindi language or Latin script. Thereby, improvements with TrEn cannot be simply due to accidental injection of source question as a hint, which may influence the answer. We report results for three values of k: 1, 3, 5.

```
Answer the following questions. Your answer must be
in the same language as the first line of the question.

Q: What is Mahatma Gandhi's national affiliation?
   महात्मा गांधी की राष्ट्रीय संबद्धता क्या है?
A: India
...

Q: ஷிகோதா ஃபீல்ட்ஸ் இன் தொழில் என்ன?
   शकोथा फील्डस का पेशा क्या है?
A:
```

```
Answer the following questions. You must first translate
the question into one random language and then answer
it in the original language.

Q: What is Mahatma Gandhi's national affiliation?
Translated question: महात्मा गांधी की राष्ट्रीय संबद्धता क्या है?
A: India
...

Q: ஷிகோதா ஃபீல்ட்ஸ் இன் தொழில் என்ன?
Translated question:
A:
```

(a) Translation Ensemble (TrEn) with k=1.  (b) Translate then Answer (TTA) with k=1.

Figure 5: Prompt templates for two input ensembling approaches. k is the number of translations presented or generated. Please refer Section 4.1.2.

**Ensembling through generation.** A model could ignore the multiple translations of TrEn-k and simply answer the first question. We can better influence the model if we redefine the task to generate translations first and then answer. We refer to the approach as **Translate-then-Answer (TTA)**. We further label the approach with TTA-k when we require the model to generate k translations before answering. Figure 5 (b) illustrates TTA-k for k=1. We report results for two values of k: 1, 3.

**Results.** We skip reporting results on MMLU (with mixup) in this section because each question in the dataset contains mixed-up language. As a result, the dataset is not suitable for monolingual translations of our ablations. Instead, we report results on Multiloko dataset in Appendix H. The dataset is described in Appendix E. Trend in our results from Table 1 generalized well to the dataset.

Unlike Section 4.1.2, our ablations TrEn and TTA yield only one response, therefore we can directly evaluate for correctness and quantify transfer using transfer scores as defined in Goldman et al. (2025). The score is defined as below.

$A_{q,l} \triangleq$ event that a question q is correctly answered in both Source and Target language: $l$

transfer-score $\triangleq \mathbb{E}_{q,l}[A_{q,l}]$

Higher values of the score indicate better overall performance and transfer. The score is 100 only if Source and Target accuracy are perfect, therefore sub-perfect score need not indicate high cross-lingual gap. In Table 1, we show the transfer score for our ablations on various models.

We observe from Table 1: (1) a consistent improvement in transfer scores from TrEn-1 to TrEn-5. (2) As expected, TTA-1 performs even better and has consistently good transfer scores with no consistent improvement from TTA-1 to TTA-3. TTA is not effective on some models like GPT-5-mini and Deepseek because they failed to follow the instruction.

|  | G-2.5-Flash | G-2.5-Pro | GPT-5-mini | GPT-5 | Deepseek | Gem-3-27B |
|---|---|---|---|---|---|---|
| Baseline | 30.7 | 37.2 | 19.1 | 35.4 | 18.0 | 9.6 |
| TrEn-1 | 32.8 | 39.2 | 23.4 | 37.6 | 19.9 | 10.3 |
| TrEn-3 | 33.7 | 40.7 | 24.2 | 38.0 | 19.5 | 10.7 |
| TrEn-5 | 36.0 | 40.6 | 22.6 | 39.3 | 18.8 | 11.5 |
| TTA-1 | 37.8 | 49.3 | 22.3 | 49.1 | 25.1 | 14.9 |
| TTA-3 | 40.6 | 48.7 | 26.0 | 46.6 | 23.6 | 11.4 |

Table 1: Source-Target transfer scores for ECLeKTic. Higher score indicate better Source-Target transfer and better overall performance. TTA-1 (highlighted) has consistently good performance. Please refer Section 4.1.2. See Table 7 of Appendix for individual Source-Target accuracies.

## 4.2 Variance in Source determines Variance in Target and Cross-lingual gaps

We plot source-target agreement with increase confidence in source for ECLeKTic, MMLU (with mixup) in this section. In Appendix F, we empirically validate that increased source confidence leads to increased target confidence. Confidence is the probability of mode as defined in Section 2.2 and Proposition 3: $\Pr(y_s^{mode})$. We estimate the confidence in practice as the relative frequency of the mode (from ten responses).

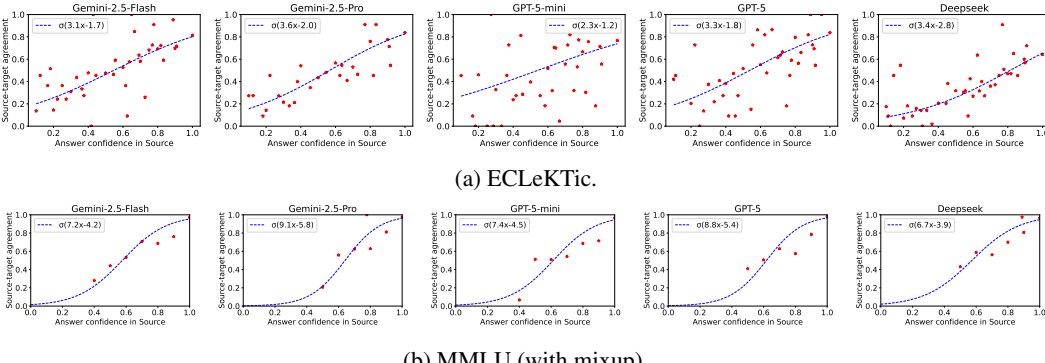

(a) ECLeKTic.

(b) MMLU (with mixup).

Figure 6: Cross-lingual gaps diminishes with reduced variance in source language. Answer confidence is defined in Section 2.2. High confidence in source leads to high confidence in target (Proposition 3), which should lead to improved agreement if there is no source-to-target bias. Section 4.2.

Since the responses in ECLeKTic are free-form, we use an LLM to summarize the multiple values into unique values and their frequency, please see Appendix M.2 for the prompt. We further use an LLM to rate if mode in source and target are matching, i.e., $y_s^{mode} = y_t^{mode}$, please see Appendix M.1 for the prompt. In Figure 6, we plot the average source-target agreement as the confidence in source increased. We observe consistent improvement in cross-lingual agreement as the source confidence improved, which confirms the trend predicted by Section 2.2 and Proposition 3.

In Appendix G, we replicate the results with Year-ECLeKTic, where we do not need an LLM to summarize responses since the responses are expected to be year. We observe that the trend comes out even more clean on Year-ECLeKTic.

## 5 RELATED WORK

**Multilingual LLMs.** Significant past work focused on understanding LLM's surprisingly effective performance on sparsely represented languages. Brinkmann et al. (2025) argued the existence of sparse multilingual features bridging the gap between the various languages. Schäfer et al. (2024) demonstrated through a toy setting the need for imbalance in the language representation in the training data for comparable performance across all languages. In practice, models trained on largely imbalanced data (like Llama-3 and Gemini) perform better or the same as the models (such as Aya 23 (Dumas et al., 2024)) trained on datasets with balanced representation of languages.

**Unsupervised Domain Adaptation (UDA).** Cross-lingual gaps are a special case of UDA where the domain is defined by the language. UDA has a very rich literature (Kouw and Loog, 2019) where a common approach is to augment the objective to reduce source-target divergence in representations along with predictive loss on the source. Methods differed in their choice of divergence measure and optimization (Ganin et al., 2016). The classic work of Ben-David et al. (2006) gave an upper bound on target risk that depends on source risk and source-target representational divergence. Thereby, many approaches suggested expensive pretraining interventions to minimize the representational divergence including multiple recent work (Wang et al., 2024a; Liu and Niehues, 2025; Ranaldi et al., 2023; Blum et al., 2025). This line of past work reflect on the traditional understanding that misaligned representations cause cross-lingual gaps. Our analysis attributed the gaps to variance and illustrated the promise of some simple approaches for fixing the gaps.

**Representation-level interpretability analysis.** Past work on interpreting language fragility in LLMs are also relevant for our work. Fierro et al. (2024); Wang et al. (2025) argue that LLMs translate to English in intermediate representations, obtain answer and then translate them back to original language. Wang et al. (2025) rationalized cross-lingual gaps through errors in translating from the intermediate English answer to the final answer. Lu et al. (2025) further argued that there are errors in both the levels of translations and proposed to fix them through steering vectors. Few other papers on mechanisms of factual recall are also related (Ferrando et al., 2024; Yuksekgonul et al., 2023; Meng et al., 2022).

# 6 DISCUSSION

We closely analyzed the causes of performance gaps in LLMs on knowledge-intensive tasks. Much of the past work hypothesized that knowledge in parameters is localized to specific languages to explain the cross-lingual gap. We demonstrated that parametric knowledge gaps are either absent or non-dominant in LLMs. Because we can often retrieve the correct answer in any language by simply ensembling multiple responses or guiding the model to do so. Instead, we hypothesized that the gaps are due to increased response variance in target setting. We validated our hypothesis through multiple targeted ablations. In all, we argue that increased variance in target rather than parametric-knowledge fragmentation as the dominant cause of the cross-lingual gap. We hope that an improved understanding of the causes can help in guiding the mitigation efforts. We recommend fixing the gaps through post-training since fragmentation of parametric-knowledge is not the dominant cause.

**Future Work and limitations.** (1) We championed some inference-time mitigation strategies that are simple and effective, and leave mitigation through training approaches for future work. (2) Our insights may also explain cross-modal (for e.g., performance disparity between text as input vs audio as input) gaps. We leave such generalizations for future work. (3) Our analysis may only apply to the languages covered by our datasets and to languages sufficiently well-represented in the training data, and will surely not apply to unseen (by LLM) languages. (4) We demonstrated that variance of responses increases in target but it is unclear what led to it (Appendix A). Is increased variance in target a coping mechanism of LLMs to keep perplexity loss from exploding due to cross-language factual inconsistencies in pretraining data? We leave such analysis also for future work.

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

# Rethinking Cross-Lingual Gaps from a Statistical Viewpoint (Appendix)

## A WHAT DETERMINES THE VARIANCE OF RESPONSES?

The main paper argued that cross-lingual gaps are due to high variance in source, which also determines the variance in target. The factors contributing to variance in source are unclear. We present some additional related insights in this section.

**Entities are a hot-spot of cross-lingual gaps.** Figure 7 presents the results on ECLeKTic along with an illustration of our source borrowed entities in target (SBET) transformation. SBET bridged 60-70% of the cross-lingual gaps to the extent that the gaps between source and SBET are statistically insignificant (p=0.05).

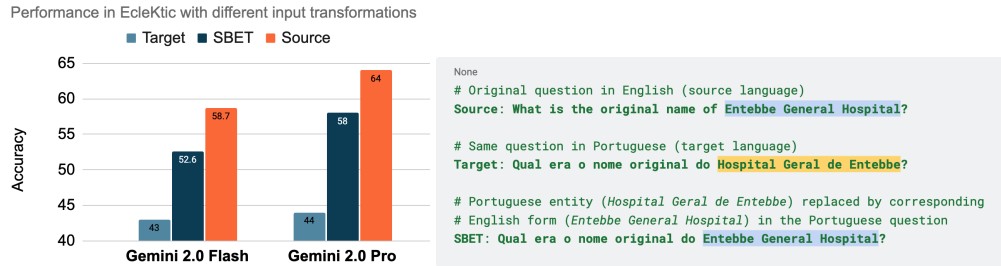

Figure 7: Illustration of Source Borrowed Entities in Target (SBET) and its performance. SBET transformation recovers 60%-70% of the cross-lingual drop suffered by Gemini 2.0 models when generalizing from Source to Target languages in ECLeKTic. Refer Section A for more details.

**(Multilingual) Popularity of Entities is uncorrelated with multilingual accuracy.** We may observe from SBET's effectiveness that the transfer of non-entity tokens is not a serious concern. It further demonstrated that entities are a hot-spot of cross-lingual factuality gaps. But it is unclear why entities in a target language are under-recognized irrespective of their mention statistics in the pretraining data. Take an example question from Hindi-sourced ECLeKTic: *Which body part of the Goddess fell at Pavagadh?* is correctly answered in almost every language even though the key entity "Pavagadh" is not mentioned in languages other than Hindi, 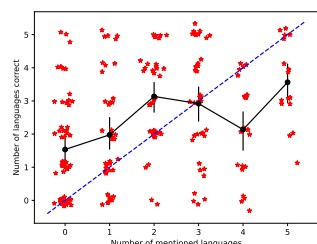 English and Korean on the web. Therefore, entities are still recognized in foreign languages even though the exact surface form was never seen before. We also illustrate the poor correlation quantitatively on ECLeKTic in the right figure. We show the number of (target) languages an entity is mentioned (on the web) and the number of languages correctly answered on the vertical axis; we only considered languages with varying scripts for the analysis: *hi, zh, ko, ja, he, en*. If not their multilingual multiplicity in pretraining data, what then determines confidence in source? Is knowledge consistency or duplication important?

## B ENSEMBLING WITH A SINGLE LLM

Our experiments in Section 4.1 prompted a single LLM multiple times and then ensemble the responses. Can we simply set the temperature to 0 for even more controlled variance? To verify the same, we repeated the ensembling experiment with `Gemini-2.5-Flash` with temperature set to 0 or 1 (the default temperature).

The results are shown in Figure 8. From the Figure we observe strange artifacts when setting the temperature to zero. Ensembling at temperature 0 is worser than ensembling at temperature 1, which

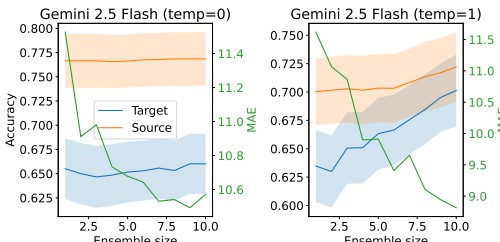

Figure 8: Replicating Figure 11 results with temperature 0 and 1 with Gemini-2.5-Flash. We observed setting temp=0 is not the same as ensembling at temp=1. Model behavior is altered with zero temperature. See Section B for more details.

is surprising. We get much better variance reduction when ensembling with default temperature than when the temperature is set to 0, observe that best MAE in left and right plots is 10.6 and 8.8 respectively.

It is unclear why we obtained better target accuracy when ensembling at non-zero temperature: 70% (at non-zero temperature) vs 65% (at zero temperature). We also note that the source accuracy is much higher when the temperature is set to zero likely because we are under-representing the variance in source responses. Also, setting temperature to 0 may internally be mapped to an unknown non-zero temperature. For instance, Deepseek-V3 maps temperature 0 to 0.3 (Deepseek, 2025).

We used the default temperature for all our experiments to avoid confounding our results with temperature related artifacts.

## C    How did we obtain embeddings of responses?

For generating response embeddings we utilize the *text-multilingual-embedding-002* (Vertex AI, 2024) API from the Vertex-AI platform. We generate embeddings for all the 10 responses for each question-language pair. Following that, we perform PCA analysis on the generated embeddings for a particular question ID for all languages. We found that often the first 2 PCA components were heavily influenced by the language script so we chose to plot 3rd and 4th component to convey semantic similarity. Finally we use kernel density plots with 5 levels to visualize the distribution or source and target language responses.

## D    Proofs

### D.1    Proof of Proposition 1

We restate the proposition for better clarity.

When the knowledge did not transfer, the probability of shared response between source and target language decreases with decreased variance.

*Proof.* **Proof sketch.** (1) We derive upper bound on the probability of shared response. (2) Then show that both the bound decreases with decreasing response variance thereby proving the statement.

**Upper bound.** We will derive the following upper bound on the probability of shared response.

$$
\Pr(\hat{y}_s = \hat{y}_t) \leq \Phi\left(\frac{\mu_0^s - \mu_1^s}{\sqrt{2(\sigma_s^2 + 2)}}\right)\left\{1 - \Phi\left(\frac{\mu_0^b - \mu_1^b}{\sqrt{2(\sigma_t^2 + 2)}}\right)\right\}
$$
$$
+ \left\{1 - \Phi\left(\frac{\mu_0^s - \mu_1^s}{\sqrt{2(\sigma_s^2 + 2)}}\right)\right\}\Phi\left(\frac{\mu_0^b - \mu_1^b}{\sqrt{2(\sigma_t^2 + 2)}}\right).
$$

Where $\mu_0^s, \mu_1^s$ are two top values of $\vec{\mu}_s$ and likewise for $\vec{\mu}_t$. $\Phi$ is the normal CDF, and m is the size of response space.

If we have two categorical distributions parameterized by $\vec{p}, \vec{q}$, the probability that samples from each distribution overlap can be computed by enumerating over the support of the distribution as shown below.

$$y_p \sim \text{Cat}(\vec{p}), y_q \sim \text{Cat}(\vec{q}).$$
$$\Pr(y_p = y_q) = \sum_k \vec{p}[k] \times \vec{q}[k] = \langle \vec{p}, \vec{q} \rangle. \tag{1}$$

Assume $\vec{p} \neq \vec{q}$ and denote by $p_a, p_b$ the top two values of $\vec{p}$, $q_c, q_d$ the top two values of $\vec{q}$ for indices $a, b, c, d$. The two top values of the expression 1 are $\max(p_a * q_a, p_c * q_c)$, $\max(p_b * q_b, p_d * q_d, \min(p_a * q_a, p_c * q_c))$ since $p_a \geq p_b \geq p_{\{k \backslash a,b\}}$ and $q_c \geq q_d \geq q_{\{k \backslash c,d\}}$. Without loss of generality, let $p_a q_a = \max(p_a q_a, p_c q_c)$. So, the two top values of the sum are $p_a q_a, \max(p_b q_b, p_d q_d, p_c q_c)$. The second highest value takes greatest value: $p_b q_c$ when b=c since $p_b \geq p_{\{k \backslash a,b\}}$ and $q_c \geq q_\bullet$. If all the probabilities are further concentrated at the two top-values, we get the following upper bound on the probability of shared response.

$$\Pr(y_p = y_q) = \langle \vec{p}, \vec{q} \rangle \leq \frac{p_a q_a + p_c q_c}{(p_a + p_c)(q_a + q_c)} \tag{2}$$

For the problem in hand, we are bounding probability of shared responses when sampling from distributions parameterized by $\vec{\mu}_s, \vec{\mu}_b$ respectively as below.

$$\vec{\epsilon} \sim \mathcal{N}(\mathbf{0}, I)$$
$$\vec{g} \sim \mathcal{G}(0, 1) \quad \text{Gumbel distribution}$$
$$y_s \sim \arg\max\{\vec{\mu}_s + \sigma_s \vec{\epsilon} + \vec{g}\}$$
$$y_b \sim \arg\max\{\vec{\mu}_b + \sigma_t \vec{\epsilon} + \vec{g}\}$$

Since we are interested in an upper bound on the probability and since probability of sharing increases with variance as $\vec{\mu}_s \neq \vec{\mu}_b$, we approximate Gumbel random variable with a random variable with even higher variance. Variance of $\mathcal{G}(0, 1) = \pi^2/6 \approx 1.6$, is approximated with $\mathcal{N}(0, 2)$.

$$y_s' \sim \arg\max\left\{\vec{\mu}_s + \sigma_s \vec{\epsilon} + \sqrt{2}\vec{\epsilon}\right\}$$
$$y_b' \sim \arg\max\left\{\vec{\mu}_b + \sigma_t \vec{\epsilon} + \sqrt{2}\vec{\epsilon}\right\}$$
$$\Pr(y_s = y_b) \leq \Pr(y_s' = y_b') \tag{3}$$

We have from Inequality 2 that we obtain an upper bound when we put all the mass of each distribution at their two highest density values and if the second highest density of one matches with the highest density of the other. We denote the two highest values of $\vec{\mu}_s$ with $\mu_0^s, \mu_1^s$ and likewise for $\mu_b$: $\mu_0^b, \mu_1^b$ in that order are the two highest values. The probability of the two levels: $\mu_0^s, \mu_1^s$ is as below.

$$\Pr(y_s = 0) = \Phi\left(\frac{\mu_0^s - \mu_1^s}{\sqrt{2(\sigma_s^2 + 2)}}\right)$$
$$\Pr(y_b = 0) = 1 - \Phi\left(\frac{\mu_0^b - \mu_1^b}{\sqrt{2(\sigma_t^2 + 2)}}\right)$$

Plugging these values into the inequality 2, we get the upper bound.

**Upper bound decreases with reduced response variance.** The upper bound is of the form: $p_1(1 - p_2) + p_2(1 - p1)$ where $p_1 = \Phi((\mu_0^s - \mu_1^s)/\sqrt{2(\sigma_s^2 + 2)})$ and similarly for $p_2$. We observe that $p_1 > 0.5, p_2 > 0.5$ since $\mu_s^0 > \mu_s^1$ and $\mu_0^b > \mu_1^b$. When response variance is reduced, both $p_1$ and $p_2$ increase but the bound decreases because $\partial p_1(1 - p_2) + p_2(1 - p_1)/\partial p_1 = 1 - 2p_1 < 0$.

$$\square$$

## D.2 PROOF OF PROPOSITION 2

We copy the statement of the proposition here for clarity.

When the target responses are unbiased, the probability of shared response between source and target language increases with decreased variance.

*Proof.* **Proof sketch.** (1) We derive an upper and lower bound on the probability of shared response, (2) we then show that both the bounds increase with reduced variance thereby proving the statement.

**Upper bound on probability of shared response.**

When the target responses are unbiased, we first show that the probability of shared response between source and target language has the following upper bound.

$$\Pr(\hat{y}_s = \hat{y}_t) \leq m \times \left\{ \Phi\left(\frac{\mu_0^s - \mu_0'^s}{\sqrt{2(\sigma_s^2 + 1)}}\right) \right\}^{m-1} \left\{ \Phi\left(\frac{\mu_0^s - \mu_0'^s}{\tau\sqrt{2(\eta\sigma_s^2 + 1)}}\right) \right\}^{m-1}$$

Where $\mu_0^s = \max \vec{\mu}_s$, $\mu_0'^s = \min \vec{\mu}_s$, $\Phi$ is the normal CDF, and m is the size of response space.

Recall that the cardinality of the response space is $m$. The probability of shared response between two distributions when enumerated over the response space is as below.

$$\vec{\epsilon} \sim \mathcal{N}(\mathbf{0}, I); \quad \vec{g} \sim \mathcal{G}(0, 1)$$
$$y_s \sim \arg\max\{\vec{\mu}_s + \sigma_s \vec{\epsilon} + \vec{g}\}$$
$$y_t \sim \arg\max\{\vec{\mu}_s + \sigma_t \vec{\epsilon} + \vec{g}\} \tag{4}$$
$$\Pr(y_s = y_t) = \sum_{k=1}^{m} \Pr(y_s = k \mid \vec{\mu}_s, \sigma_s^2 I)\Pr(y_t = k \mid \vec{\mu}_s, \sigma_t^2 I).$$

Since the mean of two distributions are matching, the probability of shared response is maximized when the variance is low. To obtain an upper bound on the shared probability, we approximate the Gumbel distribution with variance $\pi^2/6 \approx 1.6$ with $\mathcal{N}(0, 1)$. Therefore, we compute the bounds using samples from $\mathcal{N}(\vec{\mu}_s, \sigma_s^2 + 1)$ and $\mathcal{N}(\vec{\mu}_s, \sigma_t^2 + 1)$.

The probability of sampling k from the sampling distribution: $\arg\max \vec{\mu} + \sigma\vec{\epsilon}$ is the probability that $\mu_k + \sigma\epsilon_k$ is greater than any other value: $\mu_i + \sigma\epsilon_i, i \neq k$, which is $\prod_{i \neq k}\left\{\Phi(\frac{\mu_k - \mu_i}{\sqrt{2}\sigma})\right\}$.

With little more working we can derive the claimed statement as summarized below.

$$\vec{\epsilon} \sim \mathcal{N}(\mathbf{0}, I)$$
$$y_s' \sim \arg\max\{\vec{\mu}_s + \sigma_s\vec{\epsilon} + \vec{\epsilon}\}$$
$$y_t' \sim \arg\max\{\vec{\mu}_s + \sigma_t\vec{\epsilon} + \vec{\epsilon}\}$$
$$\Pr(y_s = y_y) \leq \Pr(y_s' = y_t')$$

$$= \sum_k \prod_{i \neq k} \Phi\left\{\frac{\mu_k - \mu_i}{\sqrt{2(\sigma_s^2 + 1)}}\right\} \Phi\left\{\frac{\mu_k - \mu_i}{\tau\sqrt{2(\eta\sigma_s^2 + 1)}}\right\}$$

$$\leq m \times \Phi\left\{\frac{\mu_0^s - \mu_0'^s}{\sqrt{2(\sigma_s^2 + 1)}}\right\}^{m-1} \Phi\left\{\frac{\mu_0^s - \mu_0'^s}{\tau\sqrt{2(\eta\sigma_s^2 + 1)}}\right\}^{m-1}$$

**Lower bound on probability of shared response.** We can derive the following lower bound by only considering the term where k=$\max \vec{\mu}_s$ in Equation 4. And observing that $\Phi(\frac{\mu_0^s - \mu_1^s}{\sigma}) \leq \Phi(\frac{\mu_0^s - \mu_{\backslash 0}^s}{\sigma})$.

$$\Pr(\hat{y}_s = \hat{y}_t) \geq \left\{ \Phi\left(\frac{\mu_0^s - \mu_1^s}{\sqrt{2(\sigma_s^2 + 1)}}\right) \right\}^{m-1} \left\{ \Phi\left(\frac{\mu_0^s - \mu_1^s}{\tau\sqrt{2(\eta\sigma_s^2 + 1)}}\right) \right\}^{m-1}$$

**Both the bounds increase with reduced response variance.** We can see this easily by noting that reducing response variance increases all the $\Phi$ fractions (z-scores). Therefore, reducing response variance improves source-target agreement likelihood. $\square$

### D.3 Proof of Proposition 3

We copy the proposition statement here for reference.

$$\vec{z} \sim \mathcal{N}(\vec{\mu}_s; \sigma_s^2 I), \quad \vec{z}' \sim \mathcal{N}(\vec{\mu}_s/\tau; \eta\sigma_s^2 I) \text{ when}$$

$$y_s \sim \text{Categorical}(\text{softmax}(\vec{z})), \quad y_t \sim \text{Categorical}(\text{softmax}(\vec{z}'))$$

$$y_s^{mode} = \arg\max_k \Pr(y_s = k), \quad y_t^{mode} \sim \arg\max_k \Pr(y_t = k)$$

We have the following lower bound on the probability of the mode of responses sampled from source and target.

$$\Pr(y_s^{mode}) \geq \left\{ \Phi\left(\frac{\mu_0 - \mu_1}{\sqrt{2(\sigma_s^2 + 2)}}\right) \right\}^{m-1} ; \qquad \Pr(y_t^{mode}) \geq \left\{ \Phi\left(\frac{\mu_0 - \mu_1}{\tau\sqrt{2(\eta\sigma_s^2 + 2)}}\right) \right\}^{m-1}.$$

Where m is the size of the response space and $\mu_0, \mu_1$ are the top two values of $\vec{\mu}_s$

*Proof.* The sampling of both $y_s, y_t$ can be alternately be described as below.

$$\vec{\epsilon} \sim \mathcal{N}(\mathbf{0}, I)$$
$$\vec{g} \sim \text{Gumbel}(0, 1)$$
$$y_s \sim \arg\max\{\vec{\mu}_s + \sigma_s\vec{\epsilon} + \vec{g}\}$$
$$y_t \sim \arg\max\{\vec{\mu}_s/\tau + \eta\sigma_s^2\vec{\epsilon} + \vec{g}\}$$

As we are interested in a lower bound on the probability of the mode, we will approximate Gumbel noise with Gaussian noise with higher variance like in the proof of Proposition 1. Then we have: $y_s \sim \mathcal{N}(\mu_s, \sqrt{\sigma_s^2 + 2})$. The probability of mode then is the probability of drawing the highest logit: $\mu_0$.

$$\Pr(y_s^{mode}) = \prod_{i \neq 0} \Phi\left(\frac{\mu_0 - \mu_i}{\sqrt{2(\sigma_s^2 + 2)}}\right) \geq \left\{ \Phi\left(\frac{\mu_0 - \mu_1}{\sqrt{2(\sigma_s^2 + 2)}}\right) \right\}^{m-1}.$$

The bound follows similarly for $y_t^{mode}$. □

## E  Dataset details

**ECLeKTic** (Goldman et al., 2025) dataset constitutes factual questions from Wikipedia pages that exist only in single language. Single language Wikipedia pages are used as a proxy for content that is unavailable or unpopular in other languages. Therefore, the benchmark proposed to validate the cross-lingual knowledge transfer on questions translated from the original language. Questions that are in the same language as the passage that contained the fact in pretraining data define the *source* split. While the questions in any other language make up the *target* split. It contains around 5500 examples and spans 12 languages.

**MMLU (with mixup)** (Chua et al., 2024b) dataset alters MMLU (Hendrycks et al., 2020) to probe LLM's language generalization. MMLU is a multiple choice dataset spanning multiple tasks. MMLU (with mixup) proposes to mixup the question by replacing the options with translations into random languages. Original questions make up the source split because examples with shared language for both question and options are likely in-distribution with pretraining. Likewise, questions with language mixed options make up the target split. We subsample around 2000 examples and this spans five languages. Figure 9 shows an example.

**MultiLoKo** (Hupkes and Bogoychev, 2025) is a recently released benchmark for multilingual evaluation of LLMs covering 31 languages. MultiLoKo consists of three partitions: a main partition consisting of 500 questions per language, separately sourced to be locally relevant to the specific language, and two translated partitions, containing human-authored translations from 30 non-English languages to English and vice versa. For our use-case we use the *dev-split* which is publicly available consisting of 250 questions per language along with human-authored translations.

**Languages:** ECLeKTic dataset spans twelve languages: German (de), Chinese (zh), Portuguese (pt), Spanish (es), Hindi (hi), Italian (it), Indonesian (id), Hebrew (he), Japanese (ja), French (fr), English (en), and Korean (ko). MMLU (with mixup), on the other hand, covers five languages: English (en), French (fr), German (de), Spanish (es), and Italian (it).

## F   VARIANCE IN SOURCE DETERMINES THE VARIANCE IN TARGET

In this section, we present results that support the text in Section 4.2.

Figure 10 shows average confidence in target with that of source for both the datasets.

## G   RESULTS ON YEAR-ECLeKTIC

In Figure 11, we reproduce the results on Year-ECLeKTic from Section 4.1 and Figure 4. We observe similar but cleaner trend on Year-ECLeKTic of decreasing Mean Absolute Error (MAE) between the averaged source and target answers. We also observe that the target accuracy increased with ensemble size such that the source-target accuracy differences at the extreme right are statistically insignificant for 3 of 4 models.

In Figure 12, we reproduce the results on Year-ECLeKTic from Section 4.2 and Figure 6. The figure more cleanly demonstrates the trend of increasing agreement between source and target as the confidence in source increases.

## H   RESULTS ON MULTILoKo

We also evaluate our Input Ensembling strategies as mentioned in Section 4.1.2 on MultiLoKo dataset Hupkes and Bogoychev (2025). Since our evaluation strategy required multiple translations of the question in up to 5 languages, we decided to use only the English division of the MultiLoKo dataset as it contains translations in 30 languages, while for other languages translations are only available in English language which restricts their evaluation. As observed in Table 2, following the trend as seen on ECLeKTic dataset in Table 1, we observe improvements from TrEn-1 to TrEn-5. And TTA-3 gives the best performance for most models. We noticed that some of these artifacts can also be influenced due to the input system prompt given to a specific model, hence the noise in trends.

|          | G-2.5-Flash | G-2.5-Pro | GPT-5-mini | GPT-5 | Deepseek |
|----------|-------------|-----------|------------|-------|----------|
| Baseline | 54.3        | 60.1      | 48.4       | 65.2  | 29.8     |
| TrEn-1   | 55.5        | 62.7      | 51.4       | 68.7  | 32.8     |
| TrEn-3   | 56.3        | 61.8      | 50.2       | 67.1  | 32.7     |
| TrEn-5   | 60.0        | 63.3      | 49.4       | 68.5  | 33.3     |
| TTA-1    | 57.2        | 68.2      | 48.8       | 70.6  | 25.8     |
| TTA-3    | 59.4        | 68.7      | 49.0       | 73.0  | 16.3     |

Table 2: Source-Target transfer scores for MultiLoKo. Higher score indicate better Source-Target transfer and better overall performance. TTA-3 (highlighted) has consistently good performance. Deepseek did not fare well with TTA because it often misinterpreted the instruction. Refer Section H.

> **Source:** What is the capital of France? (A) Paris (B) Rome (C) Berlin (D) Madrid
>
> **Target (with mixup):** What is the capital of France? (A) Paris (B) 로마 (C) Berlin (D) マドリード (Options B and D translated to Korean and Japanese, respectively)

Figure 9: An example from the MMLU (with mixup) dataset.

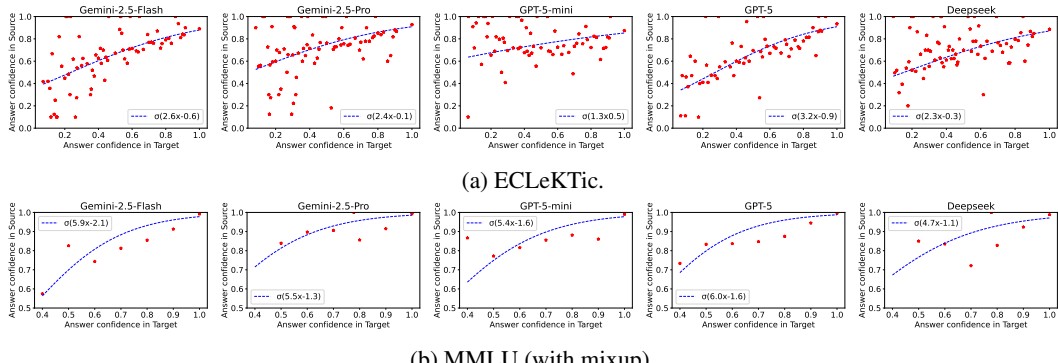

(a) ECLeKTic.

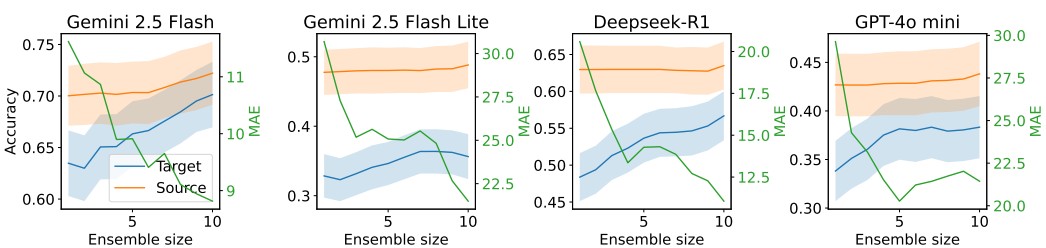

(b) MMLU (with mixup).

Figure 10: Empirical validation that high confidence in source leads to high confidence in target (Proposition 3. Answer confidence is defined in Section 2.2. The figure supports Section 4.2 of main text. Refer Section F for more details.

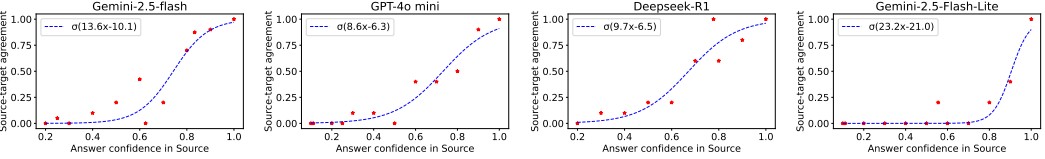

Figure 11: Reproducing results from Figure 4 on Year-ECLeKTic.

## I FINE-TUNING EXPERIMENTS

Based on our analysis in Section 2.2, we expect improved cross-lingual gaps when source accuracy is improved through overfitting. To verify the same, we finetuned a DeepSeek-R1-0528-Qwen3-8B model for 200 epochs on the source split of Year-ECLeKTic. If we succeed on overfitting the source, we must have seen improvements in target accuracy as well although target split is not part of training.

Results are shown in Figure 13. We see the predicted trend partially in the plot but we found it hard to make the model overfit on random facts from ECLeKTic. As a result, we do not have a conclusive evidence of the question we attempted to validate: *improving source alone also improves target?*

## J POTENTIAL MECHANISMS OF TARGET VARIANCE.

In Sections 2, 4.1, we argued with evidence that cross-lingual gaps are due to increased variance in target. In this section, we postulate underlying mechanisms that may lead to increased variance in target.

Figure 12: Reproducing results from Figure 6 on Year-ECLeKTic.

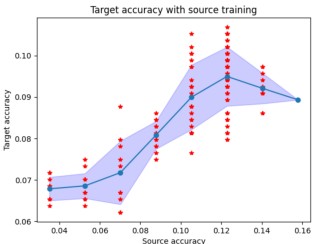

Figure 13: Performance of DeepSeek model, observing improved source target agreement with increase in source accuracy. Refer Section I.

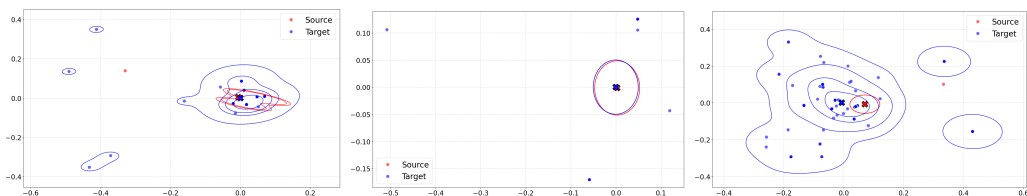

(a) According to Benjamin Mazar and Yohanan Aharoni, to which tribe did the city of Dor belong?

(b) What was the name of the University of Detroit's halfback in the 1929 football season?

(c) Alongside which council does the Israeli Coastal Environment Preservation Committee operate?

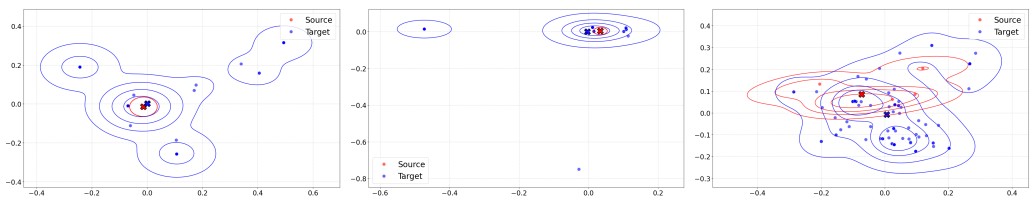

(d) Who was the first Grand Commander of Eastern Wu?

(e) Who designed the Olivetti DL typewriter?

(f) What colors do the members of the Landsmannschaft Saxonia wear?

Figure 14: Additional results from Figure 1b on ECLeKTic.

Let $\vec{r}_s, \vec{r}_t$ represent the representations of an example in source and target. The representations are expected to contain a language-agnostic and language-specific components, which we assume linearly compose the final representation, i.e., $\vec{r}_s = \vec{r} + \vec{r}_l, \vec{r}_t = \vec{r} + \vec{r}_{l'}, l \neq l'$. We use a linear model to inspire the effects of language component, thereby the response is $\vec{w}^T \vec{r}_\bullet$ for a parameter vector $\vec{w}$. The source-target responses diverge due to the language component: $\vec{w}^T \vec{r}_t = \vec{w}^T \vec{r}_s + \vec{w}^T (\vec{r}_{l'} - \vec{r}_l)$. Therefore, the target response is source response with a residual term that is dependent on language divergence.

We further assume the parameter posterior for $w$ as Gaussian that is parameterized as $\mathcal{N}(\vec{\mu}, \sigma^2 I)$. If the parameters are trained on sufficiently multilingual data, we may expect the mean to be orthogonal to the span of language variation $\vec{\mu}^T (\vec{r}_l - \vec{r}_{l'}) = 0 \quad \forall l, l'$. Finally with some working, the response distribution in source is $\mathcal{N}(\vec{\mu}^T \vec{r}_s, \sigma^2 \|\vec{r}_s\|^2)$ and target is $\mathcal{N}(\vec{\mu}^T \vec{r}_s, \sigma^2 \|\vec{r}_s\|^2 + \sigma^2 \|\vec{r}_l - \vec{r}_{l'}\|^2)$. Therefore, an additional noise in out-of-distribution target emerged due to language component of representations.

## K  DO OUR INSIGHTS FROM SECTION 4.1 HOLD FOR ANY LANGUAGES?

In this section, we investigate if our analysis and insights hold at a finer level of cross-lingual transfer: high-to-low resource and vice-versa. We picked two languages from ECLeKTic: en, zh as high-resource and two other languages: he, id as low-resource. We followed the statistics for Wikipedia pages mentioned at *WikiSources* to ascertain high and low resource languages..

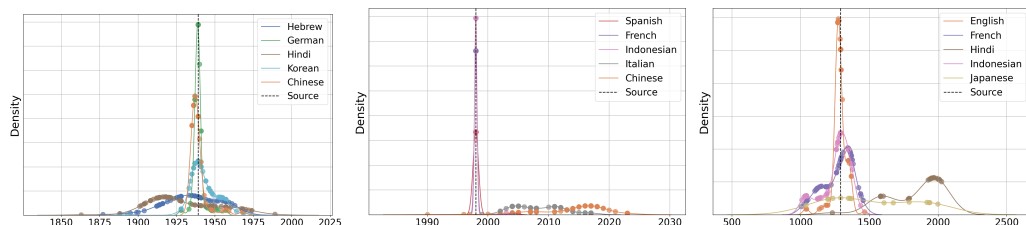

(a) In which year was Siegfried Loyda born?

(b) In which year was the first edition of the Rock Basement festival held?

(c) In what year was the Sarwadharma Inscription issued?

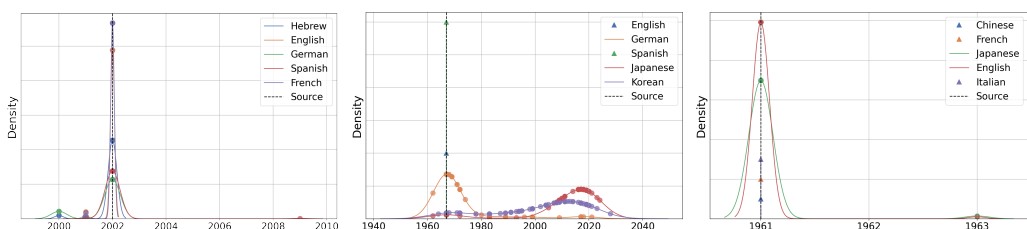

(d) In which year were the "Girotondi" movements born?

(e) In what year was the film "Three Days and a Child" released with Judith Soleh?

(f) When was the Indian Institute of Technology, Delhi established?

Figure 15: Additional results from Figure 1c on ECLeKTic.

We replicate our main findings: Figure 4, Table 1 for all combinations of high and low resource languages. All the results are consistent with the analysis in Section 4.1 with the only difference on the evaluation subset used for aggregation. For instance, high→low analysis considers only the examples that originated in a high resource language: en or zh and being queried in a low resource language: he or id. Please refer to Figures 16, 17, 18, 19 and 20. and Tables 3, 4, 5 and 6.

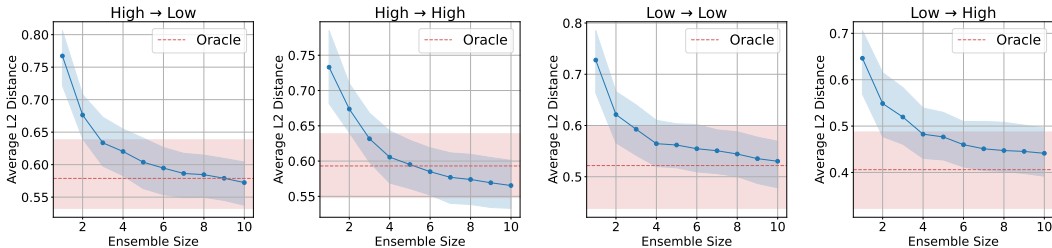

Figure 16: Reproducing results from Figure 4 on ECLeKTic for Gemini 2.5 Flash.

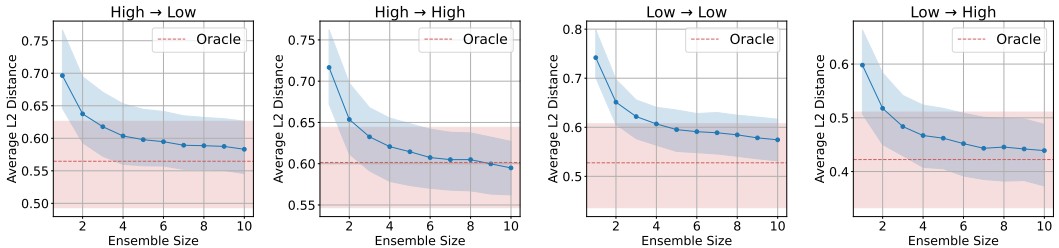

Figure 17: Reproducing results from Figure 4 on ECLeKTic for Gemini 2.5 Pro.

.

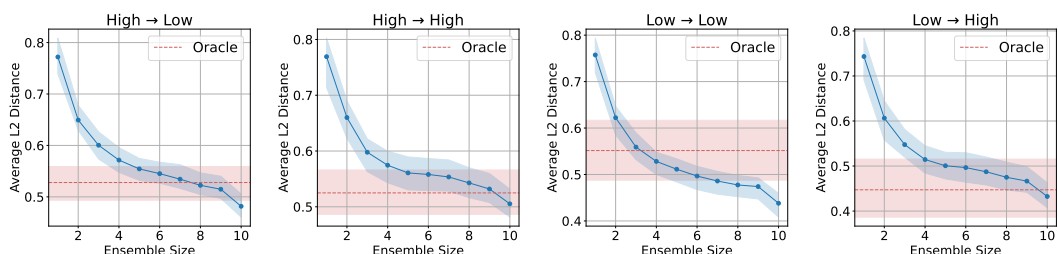

Figure 18: Reproducing results from Figure 4 on ECLeKTic for GPT-5 mini.

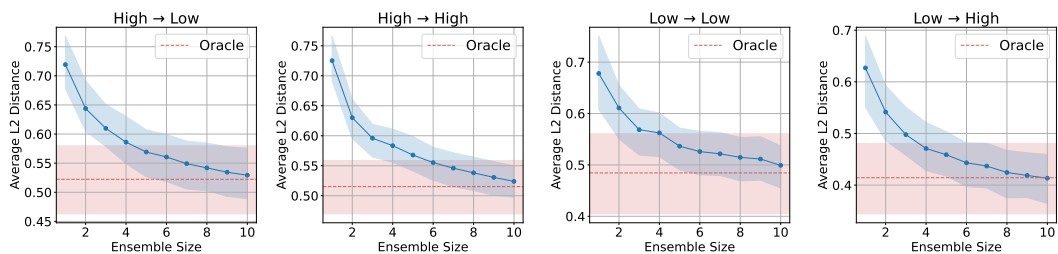

Figure 19: Reproducing results from Figure 4 on ECLeKTic for GPT-5.

## L  DO OUR INSIGHTS FROM SECTION 4.2 HOLD AT A FINER LEVEL OF CROSS-LINGUAL TRANSFER?

In this section we further experiment and solidify our findings from Section 4.2. We plot trend of source-target agreement with source confidence for 4 combinations with source and target language either being high resource or low resource. Here we use the full-set of ECLeKTic dataset and identified English, German, French, Spanish, Chinese and Italian as high resource languages and Portugese, Japanese, Korean, Hebrew, Hindi and Indonesian as low resource languages. Our findings confirm that the trend still holds for different sets of language pairs. Please refer Figure 21.

## M  MISCELLANEOUS

### M.1  RESPONSE MATCHING PROMPT

Please find the full prompt used for matching strings potentially different languages in `response_matching_prompt.txt` in supplementary material.

### M.2  RESPONSE SUMMARIZER PROMPT

Please find the full prompt used for summarizing a list of strings into their unique values and counts in `response_summarizer_prompt.txt` in supplementary material.

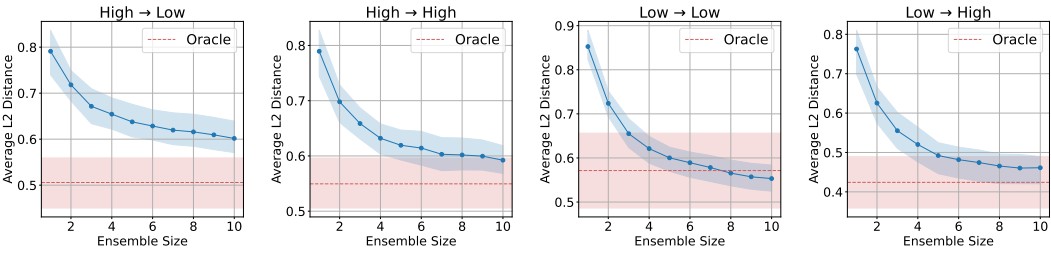

Figure 20: Reproducing results from Figure 4 on ECLeKTic for DeepSeek.

|          | G-2.5-Flash | G-2.5-Pro | GPT-5-mini | GPT-5 | Deepseek | Gem-3-27B |
|----------|-------------|-----------|------------|-------|----------|-----------|
| Baseline | 38.7        | 43.7      | 28.6       | 45.3  | 23.4     | 13.0      |
| TrEn-1   | 44.2        | 51.9      | 31.2       | 48.3  | 34.1     | 15.6      |
| TrEn-3   | 42.0        | 52.4      | 32.5       | 49.7  | 27.3     | 14.3      |
| TrEn-5   | 48.1        | 51.5      | 35.1       | 50.9  | 30.7     | 15.6      |
| TTA-1    | 43.7        | 51.1      | 34.2       | 61.0  | 38.5     | 20.3      |
| TTA-3    | 55.4        | 49.4      | 35.1       | 61.7  | 31.2     | 17.7      |

Table 3: Source-Target transfer scores for ECLeKTic for High resource language to Low resource language.

|          | G-2.5-Flash | G-2.5-Pro | GPT-5-mini | GPT-5 | Deepseek | Gem-3-27B |
|----------|-------------|-----------|------------|-------|----------|-----------|
| Baseline | 39.9        | 46.8      | 28.6       | 45.8  | 27.3     | 14.3      |
| TrEn-1   | 42.9        | 51.9      | 29.9       | 48.0  | 25.6     | 14.3      |
| TrEn-3   | 42.9        | 50.6      | 32.5       | 50.9  | 32.6     | 15.6      |
| TrEn-5   | 51.9        | 54.5      | 36.4       | 50.9  | 27.9     | 18.2      |
| TTA-1    | 40.3        | 49.4      | 36.4       | 63.4  | 37.7     | 18.2      |
| TTA-3    | 53.2        | 48.1      | 37.7       | 60.0  | 28.6     | 15.6      |

Table 4: Source-Target transfer scores for ECLeKTic for High resource language to High resource language.

## M.3 AUTOCHECKER PROMPT

Please find the full prompt used for autochecking if a response matched the reference in `autocheker_prompt.txt` attached in supplementary material. The provided autochecker prompt is for the response: *A: L'ordre de Santiago.* and reference: *ordre de santiago*.

## M.4 CHI-SQUARED DISTANCE

$$\text{Chi-squared}(p,q) = \sum_{x \text{ s.t. } p(x)+q(x)>0} \frac{(p(x)-q(x))^2}{p(x)+q(x)}$$

## M.5 PI ESTIMATION FOR MMLU (WITH MIXUP)

We provide additional details for Section 4.1.1. We estimate $\pi$ for MMLU (with mixup) as one minus fraction of examples with best answer mismatch. We use a soft score for mismatch as described next. Imagine there are only two options with source response distribution estimated as $p_0, p_1$ and target as $q_0, q_1$. The mismatch score for this examples is $|p_i - q_j|$ where $i, j$ correspond to best responses in source and target respectively, i.e., $i = \arg\max p_k, j = \arg\max q_k$. Therefore $\pi$ over the entire dataset is as below.

$$p_i^{(n)} = \arg\max p_k^{(n)}; q_j^{(n)} = \arg\max q_k^{(n)},$$

$$\pi = \frac{\sum_n |p_i^{(n)} - q_j^{(n)}|}{N}.$$

|          | G-2.5-Flash | G-2.5-Pro | GPT-5-mini | GPT-5 | Deepseek | Gem-3-27B |
|----------|-------------|-----------|------------|-------|----------|-----------|
| Baseline | 28.7        | 36.6      | 20.2       | 38.5  | 13.1     | 8.5       |
| TrEn-1   | 27.7        | 33.3      | 18.8       | 28.8  | 13.4     | 10.3      |
| TrEn-3   | 29.1        | 36.6      | 22.1       | 30.6  | 13.7     | 11.3      |
| TrEn-5   | 34.7        | 39.4      | 18.3       | 36.6  | 13.7     | 12.7      |
| TTA-1    | 34.3        | 44.1      | 23.5       | 41.9  | 22.1     | 10.8      |
| TTA-3    | 35.2        | 43.2      | 28.6       | 35.3  | 18.3     | 9.4       |

Table 5: Source-Target transfer scores for ECLeKTic for Low resource language to High resource language.

|          | G-2.5-Flash | G-2.5-Pro | GPT-5-mini | GPT-5 | Deepseek | Gem-3-27B |
|----------|-------------|-----------|------------|-------|----------|-----------|
| Baseline | 29.1        | 37.8      | 26.8       | 46.6  | 8.5      | 8.5       |
| TrEn-1   | 20.7        | 23.2      | 14.6       | 26.4  | 10.2     | 8.5       |
| TrEn-3   | 25.6        | 28.0      | 20.7       | 25.0  | 8.3      | 8.5       |
| TrEn-5   | 26.8        | 36.6      | 14.6       | 31.0  | 12.5     | 11.0      |
| TTA-1    | 31.7        | 40.2      | 28.0       | 41.7  | 22.0     | 7.3       |
| TTA-3    | 32.9        | 39.0      | 37.8       | 35.0  | 15.9     | 6.1       |

Table 6: Source-Target transfer scores for ECLeKTic for Low resource language to Low resource language.

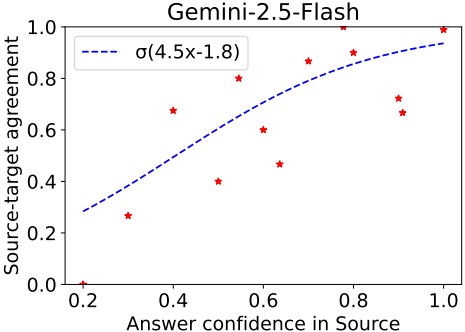

(a) High Resource Source Language, High Resource Target Language

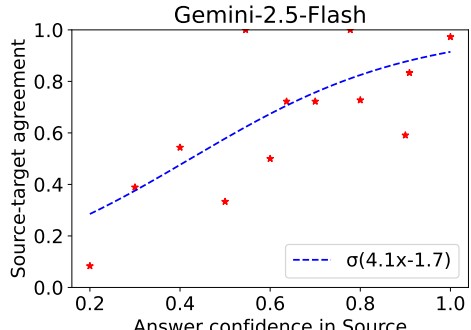

(b) High Resource Source Language, Low Resource Target Language

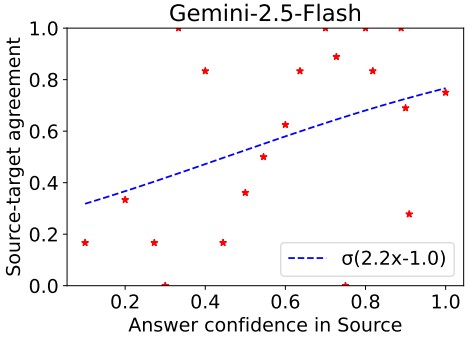

(c) Low Resource Source Language, High Resource Target Language

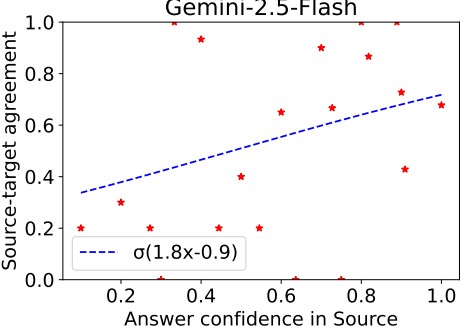

(d) Low Resource Source Language, Low Resource Target Language

Figure 21: Additional results from Section 4.2 on Year-ECLeKTic.

## M.6 WHY IS PI ESTIMATOR DIFFERENT IN ECLeKTic vs MMLU (WITH MIXUP)

The $\pi$ estimation procedure described in Section M.5 is a direct estimate for fraction of examples on which bias is likely dominant. But we could not easily adopt the estimate for ECLeKTic because our text embedding model used to embed responses is sensitive to language. Therefore, we see a large divergence in averaged target and source response embeddings just because they are in different languages.

In the interest of avoiding embedding-related artifacts in our estimate, we used the fraction of examples on which ensembling improved L2 distance as a proxy to estimate $\pi$ on ECLeKTic. This is a valid estimate because as we show in Section 2, source-target divergence improves with number of examples only when the noise is unbiased, i.e., $\kappa = 1$.

## M.7 ANALYSIS OF EXAMPLES WITH BIAS

In this section, we spot-check examples that are biased even after ensembling. We find that bias among the examples we checked is due to translation errors.

In ECLeKTic dataset we observed translations errors such as in Figure 15(b) when the Chinese question was back translated into English, we observed that "Rock Basement" was translated as "Rock Cellar" which led to erroneous answers, confirmed upon passing "Rock Basement" in the question. Similarly in Figure 15(c) upon translating Japanese question back to English we find that the word "issued" is replaced by "published" leading to erroneous answers.

Similarly, in the MMLU (with mixup) dataset, we observe for the statement-based questions, where the task is to identify if a set of statements are "Right" or "Wrong", often their translations in target languages when back translated to english can have varied meanings like "bad", which take away from the semantic meaning of the answer. On performing a spot-check we observed multiple instances of such statement-based questions.

## M.8 EXTENDED RESULTS

Below we enumerate a more comprehensive evaluation from our Table 1.

|  | G-2.5-Flash | G-2.5-Pro | GPT-5 | Deepseek |
|---|---|---|---|---|
| Baseline | 51.1 (1.4), 40.5 (0.4) | 58.7 (4.3), 46.5 (1.3) | 53.0 (1.4), 45.2 (0.4 | 38.7 (4.4), 27.1 (1.2) |
| TrEn-5 | 52.4 (4.5), 44.7 (1.4) | 57.1 (4.4), 50.9 (1.3) | 52.4 (5.1), 47.8 (1.6) | 36.3 (5.6), 30.0 (1.6) |
| TTA-1 | 56.8 (4.3), 47.1 (1.4) | 66.1 (4.2), 57.5 (1.3) | 66.2 (5.5), 55.0 (1.8) | 45.6 (10.0), 27.0 (2.8) |

Table 7: We show Source-Target accuracies for best ablations from Table 1. The format is Src acc (std dev), Tgt acc (tgt dev). Some models dropped for staying within margins.

