# OpenReview forum: "Rethinking Cross-Lingual Gaps From A Statistical Viewpoint"
_ICLR.cc/2026/Conference — Submitted to ICLR 2026_

### Official Review · Reviewer_RNgW · 2025-10-30

**Soundness:** 2
**Presentation:** 2
**Contribution:** 1
**Rating:** 2
**Confidence:** 3

**Summary:**

The study aims to understand the cause for poor cross-lingual transfer performance from a bias-variance perspective. The study claims that variance in the responses in the target language causes the cross-lingual transfer gap. Formalize bias and variance in cross-lingual transfer from source and target language responses and provide experimental results using inference-time method to reduce variance in the target language.

**Strengths:**

- identifying the cause for the cross-lingual transfer gap is very relevant and interesting
- the study views the problem from a variance-bias perspective, which is novel to the best of my knowledge
- the proposed inference-time approaches exposing the model to multilingual inputs (and asking them to explicitly translate inputs) show promising results in cross-lingual transfer performance

**Weaknesses:**

- The manuscript is not well structured and hard to follow. Experimental setup and evaluation methodologies are presented in the results section. The discussion of the results is very short and relevant parts of the discussion were moved to the appendix.
- The plots in Figure 6 seem misleading: the x-axis are inconsistent, as they should all range from 0.0 to 1.0. Also, there is no description in the caption as to how the displayed variance functions were computed. While there seems to be a trend, it is not as clear as the figures may suggest in (a) and the sample size in (b) is very low
- Inference-time methods TrEn-k and TTA are not well motivated and embedded in the rest of the manuscript and discussions.
- The derived conclusions, i.e. reducing variance in the source language reduces variance in the target language, does not hold following the conclusions in Appendix I.

**Questions:**

- PCA plot in figure 1, b is underspecified: what representations are used to compute this pca plot?
- Why would you expect responses in different target languages to exhibit similar variance as responses in a single source language?
- To test the hypothesis that target language variance is proportional to source language variance, I'd suggest to analyze languages pairwise, instead of the aggregated figures and values presented in table 1 and figure 6.

---

> ### Author Response · Authors · 2025-11-24
>
> We thank the reviewer for their valuable time and comments.
>
> > The plots in Figure 6 seem misleading: the x-axis are inconsistent, as they should all range from 0.0 to 1.0. Also, there is no description in the caption as to how the displayed variance functions were computed. While there seems to be a trend, it is not as clear as the figures may suggest in (a) and the sample size in (b) is very low
>
> We thank the reviewer for the keen observation. We changed the plot in the revised draft to have a consistent X-axis range in [0, 1].
>
>
> In Figure 6, the X-axis is “Answer confidence in Source”, which is the relative frequency of mode as described in line 430.
> In Figure (a), we compute the relative frequency of mode as follows. We sample ten responses using the source prompt, use an LLM to extract the most repeating answer (mode) and compute the fraction of times the mode occurred in responses.
> In the same figure, the Y-axis is “source-target agreement”, which we compute as follows. We use an LLM to extract the mode from source and target responses. We then again use an LLM to judge if the mode from source and target are aligned.
> We described the details in lines 449-454 of our paper.
>
> The trend in Figure 6 (a) is noisy due to error rate in LLM annotation of responses. In Figure 12 (of Appendix G), we reproduced the figure for questions on Year-Eclektic (questions with numeric response) where we do not require an LLM-judge to summarize responses. We observe a much cleaner trend in Figure 12.
>
> Despite annotation and sampling error in Figure 6, we consistently observed near perfect cross-lingual agreement when confidence in source language is near perfect.
>
> > Inference-time methods TrEn-k and TTA are not well motivated and embedded in the rest of the manuscript and discussions.
>
> We clarified the motivation for TrEn and TTA in our general comment [here](https://openreview.net/forum?id=8B8CEDMRlN&noteId=0Um5OsaWs3).
>
> > The derived conclusions, i.e. reducing variance in the source language reduces variance in the target language, does not hold following the conclusions in Appendix I.
>
> In Appendix I, we attempted a fine-tuning experiment to see if target accuracy improves with increased source accuracy since we observed reducing cross-lingual gaps if the model is confident in source. In the unfinished experiments reported in Appendix I, we could not make the (Deepseek) model memorize the facts even in the source split of Eclektic. Thereby, we remarked (in Appendix I) that our results in the section were inconclusive.
>
> Therefore, we do not see Appendix I contradicting our main claims. Please let us know if we are missing something.
>
> > PCA plot in figure 1, b is underspecified: what representations are used to compute this pca plot?
>
> We used a multilingual text embedder (text-multilingual-embedding-002 from Vertex AI) to embed the responses. We mentioned more details in L97 and Appendix C of the paper.
>
> > Why would you expect responses in different target languages to exhibit similar variance as responses in a single source language?
>
> We thank the reviewer for bringing it up. We did not claim similar variance but only claimed that they are related. We may understand their relation simply through the following argument. If the cross-lingual error is unbiased, we may express the target response = source response + unbiased noise.
> When the source variance is low, we can say that the source response (signal) is very strong, which will also lead to a strong signal and low variance in target.
>
> We argued the same more formally in Section 2.2, 4.2 and Proposition 3. We are happy to clarify further if unclear.
>
> > To test the hypothesis that target language variance is proportional to source language variance, I'd suggest to analyze languages pairwise, instead of the aggregated figures and values presented in table 1 and figure 6.
>
> We thank the reviewer for their suggestion. In the revised draft, we added Appendix L to address the concern. We split the languages in Eclektic into two groups: low and high resource based on their coverage in Wikipedia. We then replicated the results from Section 4.2 for all the four combinations of source={high, low} resource, target={high, low} resource. Pairwise analysis plots were inconclusive due to low sample size. We hope analysis with low and high resource language separation will provide some clarification.
>
> We further compare in Figure 10 the confidence in source (on Y-axis) with confidence in target (on X-axis), which is different from Figure 6. Recall that in Figure 6 of Section 4.2, we plotted source-target agreement on Y-axis and confidence in source on X-axis. We hope that direct comparison of source/target confidence provides further evidence for the relation of source and target variance.
>
> We thank the reviewer again for their time. We did our best to address their concerns. We are more than happy to engage further on any outstanding questions.

---

### Official Review · Reviewer_goJz · 2025-10-31

**Soundness:** 3
**Presentation:** 2
**Contribution:** 3
**Rating:** 6
**Confidence:** 2

**Summary:**

Overall, this paper solved the problem of understanding the causes behind cross-lingual performance gaps in multilingual LLMs. This paper proposed a statistical framework based on bias–variance decomposition, hypothesizing that cross-lingual gaps arise primarily from increased variance in target languages, rather than biases. Through formal analysis and experiments on multiple benchmarks (ECLeKTic, MMLU with mixup), the authors show that ensembling and inference-time variance control significantly reduce these gaps, suggesting that knowledge itself transfers well but confidence does not.

**Strengths:**

1) Presents a novel and rigorous statistical perspective (bias–variance decomposition) on the cross-lingual gap problem.

2) Strong empirical validation using multiple benchmarks and large models (Gemini, GPT-5, DeepSeek).

3) Well-designed experiments with both response and input ensembling to test the hypothesis.

4) Clear contributions and discussion showing practical mitigation strategies that are inference-time only.

**Weaknesses:**

1) This work focuses on inference-level variance, I'm curious if this variance/bias stil exists during training.

2) The assumption that representation bias is negligible may oversimplify real-world multilingual disparities, especially for low-resource languages.

3) Experimental scope is limited to well-represented languages; findings may not generalize to unseen or underrepresented ones.

**Questions:**

See Weaknesses.

---

> ### Author Response · Authors · 2025-11-24
>
> We are grateful for the reviewer’s time and consideration of our work.
>
> > This work focuses on inference-level variance, I'm curious if this variance/bias stil exists during training.
>
> We kindly request the reviewer to please explain what they mean by train-time variance/bias.
> Bias/variance are defined for a hypothesis class and dataset, we are not sure how to interpret inference or train-time bias/variance.
>
> > The assumption that representation bias is negligible may oversimplify real-world multilingual disparities, especially for low-resource languages.
>
> We are not sure how to interpret “representation bias”.
>
> If “representation bias” = bias due to under-representation in training data. The 12-24 languages we studied in the paper are popular but are significantly under-represented in the pretraining data. Therefore, we did not assume training data under-representation of some languages is insignificant.
>
> If “representation bias” = bias in representations. We did not argue that representations are alike in different languages. We argued in Appendix J how divergence in representations across languages could lead to increased variance in target response.
>
> If “representation bias” as in bias vs variance tradeoff we analysed. We did not assume the bias is negligible, but showed evidence that it is non-dominant.
>
> > Experimental scope is limited to well-represented languages; findings may not generalize to unseen or underrepresented ones.
>
> Thanks for pointing it out. We also acknowledged the limitation in the final section of the main paper.
>
> We are happy to engage further on any outstanding concerns.

---

### Official Review · Reviewer_kN1s · 2025-10-31

**Soundness:** 2
**Presentation:** 3
**Contribution:** 2
**Rating:** 4
**Confidence:** 4

**Summary:**

The authors investigate gaps in knowledge (and therefore performance) on tasks where the only difference is the language in which the query is posed in and adopt a bias-variance framework to explain the gap.

# Method

Logits from source language modeled as RVs from a Gaussian distribution and target logit drawn from bias and variance components, linearly weighed.
From Prop 1 and 2:

"Reducing the radii (√variance) will make the average responses from source and
target agree more often only when there are no biases"

From Prop 3.:
"
1.When the source confidence is high, i.e., (μ0 −μ1)/pσ2s + 2 ≫ 1 then the target confidence must
also be high based on Proposition 3.
2.Since source and target confidence are related, we should see increasing agreement (or suppressed
cross-lingual gap) as confidence in source increases.
"
# Results

The authors test on ECLeKTic and MMLU (w/ language mixup)

To show the gap is due to variance: embedding the L2 distance between source and target languages reduces with increasing ensemble size. Similarly, authors show decreasing Chi-square distance over multiple-choice with increasing ensemble size. Similalry, the authors do a variant of ensembling i) multiple translations presented at once and ii) translated then answer; only i) improves performance.

On ECLeKTic, authors show "High confidence in source leads to high confidence in target".

**Strengths:**

1. The authors take a principled approach to investigate the variance hypothesis and show the results on frontier performance.
2. Section 4.2 and Prop 3. results are strong and align with the presentation and inference around it.

**Weaknesses:**

1. The authors skip a lot of existing strategies (train and inference time) that people have looked at [1-3, and many others]. Discussion of the proposed framework and analyzing results from these works seems critical
2. In-line with the above comment, Section 4.1 results, especially 4.1.2, the results are a) not novel; b) novelty aside, TTA-results are extremely surprising, especially with the bigger models and attributing it failure to follow instructions doesn't seem satisfactory (unless all prompt optimizations were conducted and the models still fail to do so, which is not detailed anywhere and unlikely that is happening).
3. There is more to the results in 4.1 - there is everything from in-context learning, few-shot examples, impact of language similarity (intermediate languages), etc., all considered in previous works have to be investigated, and similar results /observations seem important.
4. Another missing analysis is correlation to language maps, where the distance between languages is calculated [4]. Previous works have shown these strategies to work, and this framework can help explain some of those empirical observations.
5. Analysis around scale of models - deeper analysis across model families should strengthen the findings.


[1] Kumar, Somnath, et al. "Bridging the Language Gap: Dynamic Learning Strategies for Improving Multilingual Performance in LLMs." Proceedings of the 31st International Conference on Computational Linguistics. 2025.
[2] Agrawal, Ashish Sunil, Barah Fazili, and Preethi Jyothi. "Translation errors significantly impact low-resource languages in cross-lingual learning." arXiv preprint arXiv:2402.02080 (2024).
[3] Wang, Weixuan, et al. "Bridging the language gaps in large language models with inference-time cross-lingual intervention." arXiv preprint arXiv:2410.12462 (2024).
[4] Littell, Patrick, et al. "URIEL and lang2vec: Representing languages as typological, geographical, and phylogenetic vectors." Proceedings of the 15th Conference of the European Chapter of the Association for Computational Linguistics: Volume 2, Short Papers. 2017.

**Questions:**

Please look at the weaknesses above.

---

> ### Author Response · Authors · 2025-11-24
>
> We sincerely thank the reviewer for their time and comments.
>
> [Q1]
> > The authors skip a lot of existing strategies (train and inference time) that people have looked at [1-3, and many others]. Discussion of the proposed framework and analyzing results from these works seems critical
>
> We thank the reviewer for pointing out some related work. We would like to gently point out that we already cite [3] (Wang et.al. ‘24), please see line 465. [1, 2] (Somnath et al., Agarwal et.al.) are tangentially related to our work, and we will consider citing them. We understand that there are many inference-time fixes for cross-lingual gaps such as [3]; we intend to demonstrate that variance is the dominant cause of cross-lingual gaps, which is much easier (than biases) to fix.
>
>
> [Q2]
> > TTA-results are extremely surprising, especially with the bigger models and attributing it failure to follow instructions doesn't seem satisfactory (unless all prompt optimizations were conducted and the models still fail to do so, which is not detailed anywhere and unlikely that is happening).
>
> Please correct us if we misinterpreted. The reviewer would like to see if TTA-1 performs better also on DeepSeek and GPT-5-mini with sufficient prompt tuning or optimization. To address their concern, we report results on Deepseek and GPT-5-mini with an improved prompt.
>
> |   | Deepseek  | GPT-5-mini |
> |---|---|---|
> | Baseline | 18.0  |19.1 |
> | TrEn-5  | 18.8  | 22.6 |
> |-----|---|---|
> | TTA-1  | 25.1  | 22.3  |
> | TTA-3  | 23.6  | 26.0  |
>
> We improved instruction following on the two models by (a) using four few shot examples (instead of two), (b) emphasised more strongly on the instruction-related constraints, which did the trick and improved performance with TTA.
> As a result, we now see significant and consistent improvements with TTA across all the models including Deepseek and GPT-5-mini.
>
> For reference, the updated system instruction for TTA is below.
>
> ```text
> You are a helpful assistant. You must strictly follow the format below to answer the question. Do not skip any steps.
> Translate: First, translate the user's question into a random language (must be different from the original language).
> Answer: Then, answer the original question in the original language.
> Constraints:
> The answer must be brief and to the point (entity or short phrase only).
> Do NOT output any conversational filler, explanations, or preamble.
> Output strictly in this format:
> Translation: [Insert Translation]
> Answer: [Insert Answer]
>  ```
>
> [Q3]
> > There is more to the results in 4.1 - there is everything from in-context learning, few-shot examples, impact of language similarity (intermediate languages), etc., all considered in previous works have to be investigated, and similar results /observations seem important.
>
> TTA and TrEn ablations already draw on in-context learning and few-shot instruction following abilities of LLMs. To the best of our efforts, we did not find any past work that teased apart bias-vs-variance attribution of cross-lingual gaps, which is the focus of our work. We also did not find past work that proposed approaches with the variance-reduction rationale, and showed consistent improvements at scale. We request the reviewer to correct us if we misinterpreted their concern.
>
> [Q4]
> > Another missing analysis is correlation to language maps, where the distance between languages is calculated [4]. Previous works have shown these strategies to work, and this framework can help explain some of those empirical observations.
>
> We thank the reviewer for the suggestion. In the revised draft, we added a new section: Appendix K to analyse the generality of our insights between languages with varying scripts: English, Hebrew, Chinese. Since languages with varying scripts are farther from each other, our analysis may potentially address the reviewer’s question. In appendix K, we demonstrated that our insights hold as well for very distant (or dissimilar) languages.
>
> [Q5]
> > Analysis around scale of models - deeper analysis across model families should strengthen the findings.
>
> We would greatly appreciate it if the reviewer could elaborate because we reported consistent results on model sizes ranging from mini to Pro spanning three model families: Gemini, GPT and Deepseek.
>
> We hope that our response addressed all the concerns. We are more than happy to engage further on any outstanding concerns.

---

### Official Review · Reviewer_Z4rW · 2025-11-11

**Soundness:** 2
**Presentation:** 3
**Contribution:** 2
**Rating:** 4
**Confidence:** 3

**Summary:**

This paper studies the cross-lingual transfer by examining the output distribution. The authors argue that the source and target languages in cross-lingual transfer share the response space, and the variance in the source language transfers to the target language. As a result, if the variance in the source language is low, the cross-lingual transfer is strong.

**Strengths:**

1.	Cross-lingual transfer is a key feature of multilingual LMs, but it is not fully understood. Studies of this feature help the community design multilingual LMs to support more languages.

2.	The authors study the cross-lingual transfer in a black box and connect the sampling process with cross-lingual transfer, which is interesting.

3.	Presentation is clear.

**Weaknesses:**

1.	While the idea is interesting, I have some general concerns or questions:

- The language bias or the language modeling performance is not identical for all languages. This might be a confounding factor in the study as the entropy or the output variance is different for all languages. One actionable suggestion here, consider studying a high-resource language to another high-resource language, high-resource to low-resource,  low-resource to high-resource, and low-resource to high-resource.
- The experimental design for ECLeKTic is not clear. Throughout all the paper, I assume the authors attempt to analyze the variance in the logits. However, for ECLeKTic, the authors consider the embedding distance via an external embedding model. This is confusing as it does not support the main claim of this paper.

2.	The prompts in the experiments are not intuitive and clear.  My understanding here (via multiple reading rounds; correct me if necessary ),  you prompt ten times for each language and compute the variance across languages. What is the intuition of using TrEn-k and TTA baselines as baselines?

**Questions:**

Please refer to Weaknesses

---

> ### Author Response · Authors · 2025-11-24
>
> We thank the reviewer for their time and valuable suggestions.
>
> > However, for ECLeKTic, the authors consider the embedding distance via an external embedding model. This is confusing as it does not support the main claim of this paper.
>
> There may have been some confusion on what we are embedding. We are embedding only the final responses, so that we can compare different responses semantically despite their language and syntactic variations. Reducing distance between averaged source and target embeddings with ensemble size supports our variance-over-bias claim.
>
> > … you prompt ten times for each language and compute the variance across languages. What is the intuition of using TrEn-k and TTA baselines as baselines?
>
> That’s right. In Section 4.1.2, we prompt with the same input multiple times. We explain the intuition for TrEn and TTA in the general comment, linked [here](https://openreview.net/forum?id=8B8CEDMRlN&noteId=0Um5OsaWs3).
>
> > One actionable suggestion here, consider studying a high-resource language to another high-resource language, high-resource to low-resource, low-resource to high-resource, and low-resource to high-resource.
>
> We thank the reviewer for the suggestion, we replicated our main results in Figure 4 and Table 1 reported for all the four combinations of transfer from low/high resource languages to high/low resource languages. We present these results in Appendix K. We observed our insights generalized to all the four combinations of source={high,low}-resource, target={high, low}-resource. We see improvements with ensembling in Appendix K figures: 16, 17, 18, 19, 20, and TTA continued to improve heavily over the baseline in Tables: 3, 4, 5, 6. For more details, please see Appendix K and the second point of the “Revised draft updates” section of our general comment linked [here](https://openreview.net/forum?id=8B8CEDMRlN&noteId=0Um5OsaWs3).
>
> We hope we addressed all the concerns, we are more than happy to engage further on any outstanding questions.

---

### Author Response · Authors · 2025-11-24
**General comment**

We sincerely thank the reviewers for their time and consideration.

We are glad that the reviewers found our presentation clear (reviewers Z4rW), our approach principled (reviewers kN1s) and our analysis interesting, rigorous or novel (reviewers Z4rW, RNgW, goJz) with strong empirical validation (reviewer goJz). We address a general concern regarding motivation for input ensembles in this section. We also summarize the edits in our paper revision.

## Motivation for TrEn-k, TTA baseline
A heuristic but popular alternate approach to ensembling is averaging model response across semantically similar inputs. Averaging responses with multiple inputs is shown to be as effective or better than ensembling with one input but multiple models [5]. Ensembling with semantically similar inputs is known as test-time augmentations [1, 2, 3, 4] and was found to be effective for improving estimates of predictions, robustness and uncertainty in various tasks: image classification, segmentation, etc.

TrEn-k is inspired from test-time augmentations. We prompt the model with k+1 semantically equivalent questions and elicit a single response under the assumption that the model implicitly ensembles across different questions. TTA is a variant on TrEn that forces the model to pay attention to all the translations (inputs).

We revised the motivation in the main paper to reflect the same.

**TL;DR**: TrEn-k and TTA are inspired from test-time data augmentation. In practice, averaging with a single example but multiple models is as effective as averaging with multiple semantically equivalent examples but with a single model.

## Updates to revised draft
We made the following changes to the draft. All the text edits are marked in blue.
* Section 4.1.2 is updated with the above text to better motivate input ensembling methods: TrEn-k and TTA.
* We added Appendix K to replicate our main results of Figure 4 and Table 1 to cross-lingual transfer between high/low resource languages. We picked two languages: English, Chinese as high-resource, Indonesian and Hebrew as low-resource, and presented results for all the four combinations of transfer between high or low resource languages. All the results are in Figures 16, 17, 18, 19, 20 and Tables 3, 4, 5, 6 of Appendix. We observed our main results generalized well to any combination of language transfer. Choice of high or low resource languages was based on language coverage on Wikipedia: https://en.wikipedia.org/wiki/List_of_Wikipedias
* We added Appendix L to replicate our Section 4.2 results for cross-lingual transfer between high or low resource languages. Figure 21 presents our results for all the four combinations of high and low resource languages for source and target.
* We updated TTA numbers in Table 1 for Deepseek and GPT-5-mini based on Reviewer kN1s suggestion. We tuned the TTA prompt for better instruction following for the two models, and improved on our earlier reported number.
* We updated Figure 6(b) to represent consistent range on the X-axis as suggested by Reviewer RNgW.
* We made a few other minor edits to fix typos and for better readability.

References:
[1] Shanmugam, Divya, et al. "When and why test-time augmentation works." arXiv preprint arXiv:2011.11156 1.3 (2020): 4.
[2] Moshkov, Nikita, et al. "Test-time augmentation for deep learning-based cell segmentation on microscopy images." Scientific reports 10.1 (2020): 5068.
[3] Ayhan, Murat Seckin, and Philipp Berens. "Test-time data augmentation for estimation of heteroscedastic aleatoric uncertainty in deep neural networks." Medical Imaging with Deep Learning. 2018.
[4] Krizhevsky, Alex, Ilya Sutskever, and Geoffrey E. Hinton. "Imagenet classification with deep convolutional neural networks." Advances in neural information processing systems 25 (2012).
[5] Kimura, Masanari. "Understanding test-time augmentation." International Conference on Neural Information Processing. Cham: Springer International Publishing, 2021.

---

### Comment · Area_Chair_6nUz · 2025-11-27
**Reviewer Reminder: Author Rebuttals Available**

Dear Reviewers,

The authors have posted their rebuttals to your reviews.

Please read the authors' responses, assess whether your concerns have been addressed, and update your rating and confidence accordingly.

Your prompt attention to the rebuttals is appreciated.

Best,
AC

---

### Author Response · Authors · 2025-12-03
**To the Area Chair and Reviewers**

We sincerely thank the reviewers for their constructive engagement, which has strengthened our manuscript. We are encouraged that reviewers found our bias–variance theoretical framework to be "principled" (**kN1s**), our analysis "novel and rigorous" (**goJz**), and our presentation "clear" (**Z4rW**).

We have addressed all raised concerns in our revision. Key highlights include:

* **Robustness across Resource Levels (Addressing [Z4rW](https://openreview.net/forum?id=8B8CEDMRlN&noteId=Gqm8fgM2KZ), [RNgW](https://openreview.net/forum?id=8B8CEDMRlN&noteId=PKMM35wYOF)):**
    To address concerns regarding language disparities, we added **Appendix K and L**. We replicated our core results from **Table 1** and **Figure 4** for cross-lingual transfer between High-Resource (English, Chinese) and Low-Resource (Indonesian, Hebrew) languages for all possible combinations (High-High, Low-Low, High-Low, Low-High). Our findings confirm that our insights on variance reduction generalize robustly across all source/target resource combinations.

* **Improved TTA Performance (Addressing [kN1s](https://openreview.net/forum?id=8B8CEDMRlN&noteId=s0TJ7lF9H7)):**
    We addressed the concern regarding *Translate-to-random-language-Then-Answer* (TTA) performance on two models(GPT-5 mini and DeepSeek). We reported minor improvements on two of the six models (Table 1) and attributed it to poor instruction following. The reviewer requested prompt tuning to fix poor instruction following with TTA on two of the models. We addressed their concern by tuning the prompt to include more specific instructions and more few-shot examples. We updated **Table 1** of the revised draft with new numbers. TTA now significantly improves over baseline for all the models, and addresses the reviewer’s concern.

* **[Clarified Motivation](https://openreview.net/forum?id=8B8CEDMRlN&noteId=0Um5OsaWs3) & Method (Addressing [RNgW](https://openreview.net/forum?id=8B8CEDMRlN&noteId=PKMM35wYOF), [Z4rW](https://openreview.net/forum?id=8B8CEDMRlN&noteId=Gqm8fgM2KZ)):**
    We revised **Section 4.1.2** to clearly articulate the motivation for TrEn-k and TTA, grounding them in the established literature of test-time data augmentation. We also corrected the axis consistency in **Figure 6(b)** as suggested by RNgW. We also provide additional results from Section 4.2 on all combinations of High and Low resource languages in **Figure 21**.
* **Overall Clarifications to Reviewers:**
   * [General Comment](https://openreview.net/forum?id=8B8CEDMRlN&noteId=0Um5OsaWs3t): Clarified TTA and TrEn motivations.
   * [Z4rW](https://openreview.net/forum?id=8B8CEDMRlN&noteId=FV7IUQf7sd): We clarified embedding generation and added new results **Appendix K**.
   * [kN1s](https://openreview.net/forum?id=8B8CEDMRlN&noteId=U2ix1OOvuS):  We improved TTA results **Table 1** with improved prompting and added new analysis.
   * [goJz](https://openreview.net/forum?id=8B8CEDMRlN&noteId=gRH8TSWqUT): We addressed the potential misinterpretation of bias-variance.
   * [RNgW](https://openreview.net/forum?id=8B8CEDMRlN&noteId=PKMM35wYOF): We corrected figures, clarified methodology and key claims and added new analysis in **Appendix K and L**.

We respectfully request the Area Chair to consider these improvements in the final decision.

---

### Meta-Review · Area_Chair_R5sK · 2026-01-07

**Summary:**

The paper argues that cross-lingual performance gaps in multilingual LLMs are primarily driven by increased response variance in the target language rather than missing knowledge or representation bias. This is a principled and novel statistical framing of cross-lingual gaps, supported by consistent empirical evidence across multiple models and benchmarks. The main weakness is that parts of the empirical story still feel underspecified or under-scoped, including unclear experimental details/structure in places, limited connection to prior cross-lingual intervention literature. Overall, I think this paper sits around the borderline.

**Reviewer Concerns:**

Concerns from Z4rW and RNgW were explicitly addressed via new results. But the confounds/measurement clarity is only partially addressed and still reads like a limitation. For reviewer kN1s, some concerns have been addressed via added results, but the bigger challenge on positioning vs. prior strategies is mostly still outstanding.

**Reviewer Scores:**

I expect some reviewers may slightly raise their scores with the newly added experimental results, but their overall assessment would likely remain negative.

---

### Decision · Program_Chairs · 2026-01-26

Reject